# DIFFEOMORPHIC OPTIMIZATION

## ABSTRACT

Optimization is a challenging task due to the rugged nature of the optimization landscape and the concentration of data on a low-dimensional manifold. Our approach starts from the observation that flow and diffusion models map the data manifold to a smooth and simple base space. We thus propose to reparameterize the optimization problem in terms of these simple base-space variables. Using concepts from differential geometry, we demonstrate that this reparameterization naturally constrains optimization to the data manifold and results in a smoother optimization surface. We extend diffeomorphic optimization to matrix groups, such as $SO(3)$ and $SE(3)$, which allows us to empirically demonstrate the effectiveness of our approach in the highly relevant task of protein design.

## 1 INTRODUCTION

Machine learning data is typically concentrated on a low-dimensional manifold (Fefferman et al., 2016; Brown et al., 2022; Kiani et al., 2024). In practice, these manifolds are not known explicitly. Many objectives of interest, such as the stability of a protein or the fidelity of a generated image, are in fact defined on low-dimensional data manifolds embedded in high-dimensional spaces. Direct optimization in this space is difficult: the objective landscape is highly non-convex, the data manifold is implicit, and unconstrained updates tend to produce out-of-distribution solutions. Consequently, while generative models have been highly successful in sampling from such manifolds, their use as a foundation for subsequent optimization is less developed.

Diffusion and flow-based generative models provide a means to address this difficulty. These models learn a diffeomorphic map from a tractable base distribution to the data distribution. This transformation reparameterizes the data manifold: complex, multimodal regions in data space correspond by design to smoother and more regular regions in the base space of the prior distribution. Although this property has primarily been exploited for sampling, it also suggests a principled approach to optimization.

In this work, we propose diffeomorphic optimization - a novel method to optimize arbitrary differentiable cost functions on the data manifold. Our method harnesses the recent advances

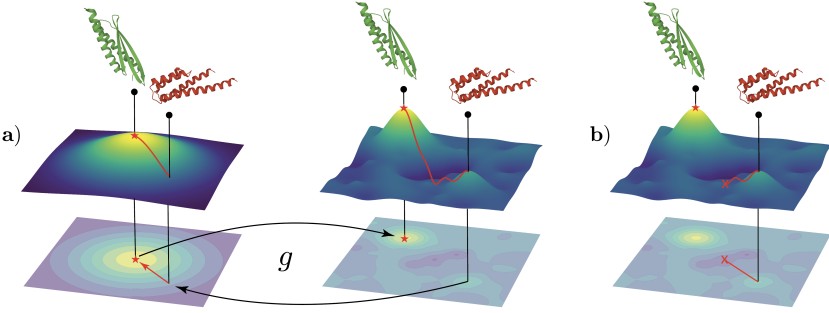

Figure 1: Optimization of protein with undesired properties (red) to a protein of desired properties (green). **Right:** Direct optimization in target space gets stuck in local minima. **Left:** Optimization in base space leads to smoother objective that allows for mode-switching.

in flow matching and diffusion models, is applicable to a wide range of tasks, and scales to high dimensional data, such as proteins with several hundreds of residues.

Diffusion and flow models learn a diffeomorphic (smooth and invertible) map $g : \mathcal{Z} \to \mathcal{X}$ that maps a base space $\mathcal{Z}$, equipped with a simple probability density $q_{\mathcal{Z}}$, to the target space $\mathcal{X}$, for example, the space of atom positions. The invertible map $g$ is parameterized by a neural network and induces a density $q_{\mathcal{X}}(x) = q_{\mathcal{Z}}(g^{-1}(x))|\det \frac{dg^{-1}}{dx}|$ on the target space $\mathcal{X}$. The training objective then aims to ensure that this pushforward density $q_{\mathcal{X}}(x)$ agrees with the data distribution to a good approximation. As a result, $g$ maps base space elements $z \in \mathcal{Z}$ to the data manifold $\mathcal{D} \subset \mathcal{X}$ with high probability.

Consider a cost function $\mathcal{L} : \mathcal{X} \to \mathbb{R}$, e.g. the energy or thermostability of a protein, that we want to optimize on the data manifold $\mathcal{D}$. Since the map $g$ is diffeomorphic, we change coordinates to the base space variables,

$$\mathcal{L} \circ g : \mathcal{Z} \to \mathbb{R}, \quad z \mapsto \mathcal{L}(g(z)) \,. \tag{1}$$

We can then perform gradient descent with learning rate $\lambda \in \mathbb{R}$ in the base space variables

$$z^{(i+1)} = z^{(i)} - \lambda \nabla_z \mathcal{L}(g(z^{(i)})), \qquad \text{for } i \in \{1, \ldots, n\} \,, \tag{2}$$

and map the final base point $z^{(n)}$ to its corresponding $x^{(n)} = g(z^{(n)})$. This parameterization has several advantages: the map $g$ is bijective and thus no information is lost. The data distribution is often highly multimodal. However, the density of the base space $q_{\mathcal{Z}}$ is chosen to be a simple unimodal distribution, such as a normal distribution $\mathcal{N}(0, 1)$. Thus, the base space reparameterization makes it easier to switch modes leading to a smoother loss landscape. Furthermore, since a well-trained $g$ maps base space samples with high probability to the data manifold, the gradient-descent optimization is equivalent to gradient descent on the data manifold, as we show in Section 3 using differential geometry. Diffeomorphic optimization is illustrated in Figure 1.

For flow matching and diffusion models, the diffeomorphic map $g : \mathcal{Z} \to \mathcal{X}$ is constructed by integrating an ordinary differential equation $\frac{dx_\tau}{d\tau} = v_\theta(x_\tau, \tau)$:

$$g(z) \equiv x_1 = z + \int_0^1 d\tau \, v_\theta(x_\tau, \tau) \,. \tag{3}$$

For flow matching, the neural network $v_\theta$ is trained to approximate the conditional ground truth vector field for a given probability path, see e.g. Holderrieth & Erives (2025) for a review. Diffusion models parameterize the vector field $v_\theta(x_\tau, \tau) = f(x_\tau, \tau) + \frac{1}{2}g^2(\tau)s_\theta(x_\tau, \tau)$ in terms of a score network $s_\theta(x_\tau, \tau)$ Song et al. (2020). Here, $f$ and $g$ are the drift and diffusion coefficient of the noising process, respectively. Diffeomorphic optimization therefore requires backpropagation through the numerical integrator used to obtain equation 3.

This task is complicated by the fact that generative models for proteins often do not sample the atom positions directly but rather the $SE(3)$ backbone frames and $SO(2)$ sidechain dihedral angles Yim et al. (2023b;a); Jing et al. (2024); Jumper et al. (2021). We will therefore derive two efficient methods to facilitate backpropagation through ODE solvers on matrix Lie groups in Section 4. The first relies on repurposing existing autograd engines for calculating the Riemannian gradient. For the second, we derive a suitable generalization of the adjoint-state method for matrix Lie groups.

We showcase diffeomorphic optimization in several numerical experiments. Specifically, we demonstrate that we can start from a given protein and then controllably change (parts of) its secondary structures using the flow matching model FrameFlow Yim et al. (2023b) as the diffeomorphic map. Using the DiffDock diffusion model Corso et al. (2023), we show that we can optimize the Vina docking score Trott & Olson (2010) for a protein-ligand pocket. Furthermore, we demonstrate that we can minimize the Rosetta energy function of a given protein in the base space of the AlphaFlow model Jing et al. (2024). This relaxation protocol leads to significantly lower energies than the state-of-the-art Rosetta Relax.

Our work coincides with a recent rise of inference computation for protein design. Practitioners often sample thousands of designs, rank, and submit only a handful for further wetlab

experiments Bennett et al. (2024); Abramson et al. (2024); Watson et al. (2023); Frey et al. (2025). The reason for this is that experimental wet lab capacity (and not sampling costs) is the main bottleneck. Furthermore, extensive sampling has been shown to lead to pronounced performance improvements, for example, for antibody-antigen coupling Abramson et al. (2024). Diffeomorphic optimization provides a more targeted approach to obtain high quality samples compared to brute-force sampling followed by ranking and selection. We therefore believe that it will become an important part of the inference compute toolbox for proteins.

Briefly summarized, our key contributions are the following. We propose diffeomorphic optimization that is applicable to any differentiable optimization objective. As we show theoretically, it automatically enforces the manifold constraint and leads to a smoother optimization landscape. To enable this, we propose two effective methods to facilitate backpropagation through matrix Lie group ODE solvers and will provide efficient PyTorch implementations for it. Experimentally, we apply diffeomorphic optimization to tasks of high practical relevance, i.e. protein ligand docking, Rosetta energy relaxation, and secondary structure modification, by combining it with state-of-the-art generative models in the field of protein generation, specifically FrameFlow, DiffDock, and AlphaFlow.

## 2 Related Works

Protein hallucination is a version of computational protein design which uses backpropagation through a folding model to its input sequence Anishchenko et al. (2021); Kosugi & Ohue (2022); Goverde et al. (2023); Pacesa et al. (2024); Cho et al. (2025). Gradient descent and normalizing flows has been explored in the explainability literature to generate counterfactual explanations Joshi et al. (2019); Dombrowski et al. (2021; 2023); Dhurandhar et al. (2018) although not for flow matching and diffusion models. Ben-Hamu et al. (2024) proposes to differentiate through flows for controlled generation. Our work builds on this reference by generalizing it to matrix Lie groups, which is of high relevance for proteins. We also provide a detailed theoretical analysis of the method and derive an efficient adjoint state method. Wang et al. (2024) explores related ideas for matrix groups in the framework of optimal control. Specifically, the authors propose a nice framework which adds an additive control to the vector field of the flow. Our approach does not use a control but rather optimizes the initial condition following Ben-Hamu et al. (2024). We discuss the relationship to this reference in more detail in Appendix B and compare in detailed numerical experiments to their approach. Liu et al. (2023) similarly relies on control variables but directly applies gradient descent to them. This reference does however not consider matrix groups. The adjoint state method on manifolds has been discussed in other works for charts Lou et al. (2020); Mathieu & Nickel (2020) or particular manifolds Rezende & Mohamed (2020); Bacchio et al. (2023); Albergo & Vanden-Eijnden (2022). Guidance is a widely used method to bias diffusion and flow-matching models towards certain desiderata is guidance. There exist various flavors of it, such as classifier-based guidance Dhariwal & Nichol (2021), classifier-free guidance Ho & Salimans (2022), and universal guidance Bansal et al. (2023). Further information can be found in the appendix B.

## 3 Diffeomorphic Optimization stays on Manifold

In this section, we analyze the diffeomorphic optimization procedure theoretically using differential geometry. In particular, we will demonstrate its relation with gradient descent on the manifold.

### 3.1 Basics Concepts of Differential Geometry

**Manifolds and coordinates:** a manifold $\mathcal{M}$ is a space that locally takes the form of $\mathbb{R}^D$, similar to the earth which can locally approximated by flat three-dimensional space. More formally, for each point $p \in \mathcal{M}$, there exists a chart $\varphi : U \to \mathbb{R}^D$ where $U$ is an open subset of $\mathcal{M}$ containing the point $p$. In this sense, the manifold is locally described by a Euclidian $D$ space. The pair $(U, \varphi)$ is called coordinate chart and the component functions $x^i$ of $\varphi(p) = (x^1(p), \cdots, x^D(p))$ are called coordinates.

**Tangent space:** the tangent space $T_p\mathcal{M}$ contains the velocity vectors $\frac{d}{dt}\gamma(t)|_{t=0}$ of curves $\gamma : \mathbb{R} \to \mathcal{M}$ with $\gamma(0) = p$. It can be shown that the tangent space $T_p\mathcal{M}$ is a $D$-dimensional vector space. Let $(U, \varphi)$ be a coordinate chart on $\mathcal{M}$ with coordinates $x$. We can then define $\varphi \circ \lambda_k(t) = (x^1(p), \ldots, x^k(p) + t, \ldots, x^D(p))$ with $k \in \{1, \ldots, D\}$. This implicitly defines curves $\lambda_k : \mathbb{R} \to \mathcal{M}$ through $p$. We denote the corresponding tangent vectors as $\frac{\partial}{\partial x^k} = \frac{d}{dt}\lambda_k(t)|_{t=0}$ and it can be shown that they form a basis of the tangent space $T_p\mathcal{M}$. The differential $dg_p : T_p\mathcal{M} \to T_{f(p)}\mathcal{N}$ for a map $g : \mathcal{M} \to \mathcal{V}$ between two manifolds $\mathcal{M}$ and $\mathcal{V}$ is a linear map between the respective tangent spaces. The curve $\gamma_v : \mathbb{R} \to \mathcal{M}$, corresponding to the vector $v \in T_p\mathcal{M}$, is mapped by the differential $dg_p$ to the curve $g \circ \gamma_v : \mathbb{R} \to \mathcal{N}$. Thus, the differential acts as $dg_p[v] = \frac{d}{dt}g \circ \gamma_v(t)|_{t=0}$.

**Riemannian metric:** A Riemannian manifold is endowed with an inner product $\langle \cdot, \cdot \rangle_{G_p} : T_p\mathcal{M} \times T_p\mathcal{M} \to \mathbb{R}$ for each tangent space, which allows us to define a notion of the length of tangent vectors. $G$ is also known as the Riemannian metric. We refer to the excellent textbook Lee (2018) for more details.

**Riemannian gradient:** The Riemannian gradient $\mathrm{grad}_p^G f \in T_p\mathcal{M}$ at point $p \in \mathcal{M}$ of a function $f : \mathcal{M} \to \mathbb{R}$ is the vector uniquely defined by the relation

$$\forall v \in T_p\mathcal{M} : \qquad\qquad \langle v, \mathrm{grad}_p^G f \rangle_{G_p} = df_p[v] \qquad\qquad (4)$$

and crucially depends on Riemannian metric $G$ of the manifold. In coordinates, the Riemannian gradient is $\widehat{\mathrm{grad}_p^G} f = \sum_j G^{ij}\partial_j f$ where $G^{ij}$ is the inverse of the metric tensor $G_{ij} = G(\frac{\partial}{\partial x^i}, \frac{\partial}{\partial x^j})$.

**Exponential map:** On a Riemannian manifold, we can further define geodesics, i.e. curves $\gamma : I \subset \mathbb{R} \to \mathcal{M}$ that have vanishing acceleration with respect to the covariant derivative induced by the Riemannian connection. For each $p \in \mathcal{M}$ and $v \in T_p\mathcal{M}$, there exists a unique geodesic $\gamma_v$ that satisfies $\gamma_v(0) = p$ and $\frac{d}{dt}\gamma_v(t)|_{t=0} = v$. The exponential map is then defined by

$$\exp_p : T_p\mathcal{M} \to \mathcal{M}, \quad v \mapsto \gamma_v(1) \,. \qquad\qquad (5)$$

Intuitively, the exponential map formalizes the notion of taking a step in the direction of the tangent vector $v$ and then finding the closest point on the manifold.

**Gradient descent on manifolds:** adding and scaling points on the manifold will not necessarily lead to points on the manifold. In contrast, such operations acting on tangent vectors do lead to other tangent vectors, since a tangent space is a vector space. This suggests a generalization of gradient descent on manifolds: we first scale the gradient of a loss $\mathcal{L} : \mathcal{M} \to \mathbb{R}$, which in particular is a tangent vector, by the learning rate $\lambda \in \mathbb{R}$. To obtain the updated point $p^{i+1} \in \mathcal{M}$, we apply the exponential map at the previous point $p^i \in \mathcal{M}$ to the rescaled gradient:

$$p_{i+1} \leftarrow \exp_{p_i}(-\lambda\,\mathrm{grad}_{p_i}^G \mathcal{L}) \,. \qquad\qquad (6)$$

In $\mathbb{R}^n$, the exponential map is $\exp_p(v) = p + v$ leading to standard gradient descent.

### 3.2 Diffeomorphic optimization is equivalent to Gradient Descent on the Data Manifold

Let $g : \mathcal{Z} \to \mathcal{D}$ be a generative model that maps its latent space $\mathcal{Z}$ to the data manifold $\mathcal{D}$. Given such a model, we can then perform gradient descent in its latent space and map the resulting point on the data manifold. We then show:

**Theorem 1.** *Let $g : \mathcal{Z} \to \mathcal{D}$ be a diffeomorphic generative model and $\mathcal{L} : \mathcal{D} \to \mathbb{R}$ is the loss. Then, up to quadratic corrections in the learning rate $\lambda$, performing gradient descent in the latent space $\mathcal{Z}$ and then mapping it go the data space $\mathcal{D}$ with $g$ is equivalent to gradient descent on the data manifold, i.e.,*

$$g(\exp_z(-\lambda\,\mathrm{grad}_z^{\tilde{G}}\mathcal{L} \circ g)) = \exp_{g(z)}(-\lambda\,\mathrm{grad}_{g(z)}^G \mathcal{L} + \mathcal{O}(\lambda^2)) \,, \qquad\qquad (7)$$

*where $G(u, v) = \tilde{G}(dg^{-1}u, dg^{-1}v)$ denotes the pushforward of the Riemannian metric $\tilde{G}$ on $\mathcal{Z}$.*

*Proof.* See Appendix C.1. □

In coordinates, the gradient in data space is given by

$$\widehat{\text{grad}}_{g(\hat{z})}\mathcal{L} = J_g \widehat{\text{grad}}_z\mathcal{L}\,, \qquad \text{with } J_g = \frac{\partial \hat{g}}{\partial \hat{z}}\,. \qquad (8)$$

We can perform a SVD decomposition of the Jacobian $J_g$. In the basis of its left singular values, the learning rate will be scaled by the corresponding singular vectors. The data manifold $\mathcal{D}$ is heavily concentrated and thus we expect low values for the singular vectors along these contracted directions. This gives us a mechanism to extract the implicitly learned tangent space from the generative diffusion or flow-matching model, which is (approximately) spanned by the remaining left-singular vectors.

## 4 BACKPROPAGATION THROUGH SE(3) ODE SOLVERS

The frame representation of proteins considers an idealized backbone geometry of its heavy atoms $[N, C_\alpha, C, O] \in \mathbb{R}^{3,4}$ determined by fixed bond lengths and angles Jumper et al. (2021); Yim et al. (2023b). A protein consists of multiple residues and for each residue $i$, the main backbone atoms can be described by a simple rotation $R_i \in SO(3)$ and translation $t_i \in \mathbb{R}^3$ of these idealized backbone coordinates, as seen in Figure 2,

$$[N_i, C_i, , (C_\alpha)_i] = T_i[N, C, C_\alpha] \qquad (9)$$

where $T_i = (R_i, t_i)$ is an element of the three-dimensional special Euclidean group $SE(3) = SO(3) \ltimes \mathbb{R}^3$. The positions of the remaining heavy atoms (backbone oxygen and sidechain carbons) can be fixed by dihedral angles. Therefore, the vector field $v_\theta$ in the ODE equation 3 takes value in the tangent space of the corresponding Lie groups $SE(3)$ and $SO(2)$. Integration on $SO(2)$ can simply be performed by standard numerical integration in $\mathbb{R}^3$ and then wrapping the result on the circle $S^1$. The case of $SE(3)$ is, however, highly non-trival. For example, the group $SE(3)$ has the product rule

$$T_1 T_2 = (R_1, t_1)(R_2, t_2) = (R_1 R_2, R_1 t_2 + t_1)\,. \qquad (10)$$

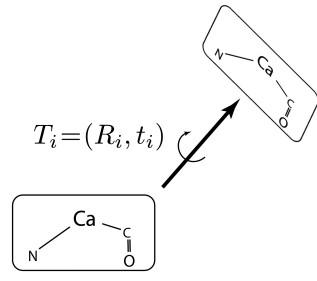

Figure 2: $SE(3)$ roto-translation of the idealized backbone positions.

The translation group $\mathbb{R}^3 \simeq \{(I, t)|t \in \mathbb{R}^3\}$, where $I$ is the unit element of $SO(3)$, is an abelian normal subgroup of $SE(3)$ since $(R, t)(I, t')(R, t)^{-1} = (I, Rt')$. Therefore $SE(3)$ is not semi-simple and there is no canonical left- and right-invariant Riemannian metric induced by the Killing form. In the generative model literature, one typically chooses the Riemannian metric to be the sum of the metrics of $SO(3)$ and $\mathbb{R}^3$ Yim et al. (2023b). This choice is right-invariant but not left-invariant[1]. For this choice, the exponential maps and Riemannian gradients for the translations and rotations decouple

$$\exp_T((\omega, v) = (\exp_R(\omega), \exp_t(v))\,, \qquad \text{grad}_T f = (\text{grad}_R f, \text{grad}_t f)\,, \qquad (11)$$

for $\omega \in \mathfrak{so}(3)$, $v \in \mathfrak{r}^3 \simeq \mathbb{R}^3$, and $f : SE(3) \to \mathbb{R}$. The explicit form of these exponential maps and gradients for both translations and rotations are:

**Translations:** since $\mathbb{R}^3$ is a vector space, it holds that

$$\exp_t(v) = t + v\,, \qquad \text{grad}_t f = \nabla_t f(t, R) \qquad (12)$$

where we use the canonical isomorphism $\mathfrak{r}^3 \simeq \mathbb{R}^3$ and $\nabla_t$ denotes the standard gradient operator $\nabla = (\partial_1, \partial_2, \partial_3)$. As a result, we recover standard gradient descent for the translational part

$$t^{i+1} = t^{(i)} - \lambda \nabla_{t^i}\mathcal{L}(t^i, R^i)\,. \qquad (13)$$

---

[1]The question of whether the metric is left- or right-invariant is a matter of convention. We choose the Riemannian metric to be induced by push-forward of the Killing form with respect to the right multiplication.

**Rotations:** as we discuss in Appendix D, the right multiplication $R_g : G \to G$, $h \mapsto hg$ induces a isomorphism between the tangent spaces $T_g G$ and the Lie algebra $\mathfrak{g} \simeq T_e G$ for any Lie group $G$. It is often easier to work with the exponential map $\exp : \mathfrak{g} \to G$ in terms of Lie algebra elements. For matrix Lie groups, such as $SO(3)$, this exponential map is simply given in terms of the matrix exponential $\exp(v) = \sum_{i=0}^{\infty} \frac{v^n}{n!}$ where $v$ is the matrix Lie algebra element. Similarly, we can uniquely associate the Riemannian gradient $\mathrm{grad}_R f$ of a function $f : SO(3) \to \mathbb{R}$ with the Lie algebra element

$$\nabla f(R) = \sum_a T^a \nabla^a f \in \mathfrak{so}(3) \,, \qquad \nabla^a f(R) = \frac{d}{dt} f(\exp(tT^a)R)\Big|_{t=0} \,, \qquad (14)$$

where $T^a$ denote the antisymmetric generators of the Lie algebra $\mathfrak{so}(3)$. Therefore, gradient descent on $SO(3)$ amounts to

$$R^{i+1} = \exp(-\lambda \nabla_{R^i} \mathcal{L}(R^i, t^i)) R^i \,. \qquad (15)$$

**Summary:** Gradient descent on $SE(3)$ can be performed by

$$T^{i+1} = \begin{bmatrix} R^{i+1} \\ t^{i+1} \end{bmatrix} = \exp_{T^i}(-\lambda \nabla_{T^i} \mathcal{L}(T^i)) = \begin{bmatrix} \exp\left(-\lambda \nabla_{R^i} \mathcal{L}(R^i, t^i)\right) R^i \\ t^{(i)} - \lambda \nabla_{t^i} \mathcal{L}(t^i, R^i) \end{bmatrix} \qquad (16)$$

In this sense, the gradient descent of the rotational and translational part decouples.

### 4.1 GRADIENTS OF **SE(3)** SOLVERS

Let $g : SE(3)^n \to SE(3)^n$ be a diffeomorphism. We restrict to $n = 1$ for notational simplicity but the results immediately generalize to $n > 1$. We use the notation $(R, t) = g(Z, z)$ with $Z \in SO(3)$ and $z \in \mathbb{R}^3$ denoting the base space variables. We can thus reparameterize the loss function $\mathcal{L} : SE(3) \to \mathbb{R}$ by

$$\mathcal{L}(g(Z, z)) \qquad (17)$$

for which we want to perform gradient descent with respect to the base variables $(Z, z) \in SE(3)$. In particular, we can consider flow matching and diffusion models for which the diffeomorphism $g$ is defined by $g(Z, z) \equiv (Z_1, z_1)$ where the right hand side is the solution of the following initial value problem on $SE(3)$:

$$dT_\tau = (dZ_\tau, dz_\tau) = (\, V_\theta(Z_\tau, z_\tau), v_\theta(Z_\tau, z_\tau)\,)\, d\tau \,,$$
$$T_0 = (Z_0, z_0) = (Z, z) \,, \qquad (18)$$

where $\tau \in [0, 1]$ and $V_\theta \in \mathfrak{so}(3)$, $v_\theta \in \mathbb{R}^3$ are parameterized by neural networks. As a result, we need to backpropagate through the numerical solver of the ODE. Specifically, we need to calculate the Riemannian gradient $\nabla_Z \mathcal{L}(g(Z, z))$ with respect to rotation $Z \in SO(3)$. In the following, we will propose two methods to do so.

**Repurposing Autograd for SO(3):** the Riemannian gradient can be expressed in terms of a simple matrix derivative:

**Theorem 2.** *The Riemannian gradient equation 14 on $SO(3)$ of a loss function $\mathcal{L} : SO(3) \to \mathbb{R}$ can be written as*

$$\nabla \mathcal{L}(R) = 2 \left[ \frac{df}{dR} R^\top \right]_A \,, \qquad (19)$$

*where $\frac{df}{dR}$ denotes the standard matrix-calculus gradient with respect to the matrix $R \in SO(3)$. Furthermore, we denote the antisymmetric component of a matrix $M$ by $[M]_A = \frac{1}{2}(M - M^\top)$.*

*Proof.* See Appendix C.2. □

This suggests a straightforward way to calculate Riemannian gradients with modern autograd frameworks. Specifically, one wraps any $SO(3)$-valued leaf variable in an autograd method acting as the identity in the forward pass but performing the multiplication equation 19 in its backward where $\frac{df}{dR}$ is the output gradient. The gradient of this variable will correspond to the antisymmetric Riemannian gradient matrix equation 14, see Listing 1 for a pytorch implementation of this. We emphasize that this approach is completely general and ensures that the Riemannian gradient is seamlessly integrated in existing autograd functionality.

```
import torch
from scipy.spatial.transform import Rotation

def so3tensor(*args, **kwargs):
    x = torch.tensor(*args, **kwargs)

    def so3_backward_hook(grad):
        so3_grad = grad @ x.transpose(-1, -2)
        return so3_grad - so3_grad.transpose(-1, -2)

    if x.requires_grad:
        x.register_hook(so3_backward_hook)
    return x

x = so3tensor(Rotation.random(1).as_matrix(), requires_grad=True)
y = so3tensor(Rotation.random(1).as_matrix())

loss = (x - y).pow(2).sum()
so3_grad = torch.autograd.grad(loss, x)[0]
print(so3_grad)

>>tensor([[[ 0.0000,  0.8490,  0.1187],
           [-0.8490,  0.0000, -2.2494],
           [-0.1187,  2.2494,  0.0000]]], dtype=torch.float64)
```

Listing 1: Repurposing Autograd for $SO(3)$ manifolds. Adding the `so3_backward_hook` repurposes autograd to compute Riemannian gradients for rotation matrices.

In particular, we can facilitate backpropagation through ODE solvers on $SO(3)$ by combining this repurposing trick with standard activation checkpointing at intermediate points of the integration trajectory. This allows us to limit the memory footprint of the computational graph to the desired degree.

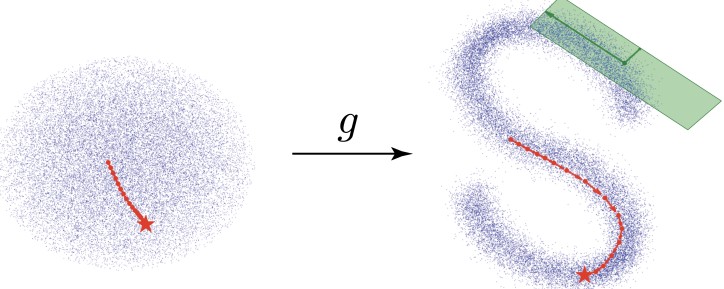

Figure 3: Diffeomorphic Optimization on $SO(3)$: left hand side visualizes the gradient descent trajectory in the base space $\mathcal{Z}$. Right hand side visualizes the same trajectory when mapped to the target space $\mathcal{X}$. Diffeomorphic optimization clearly stays on manifold. The green plane is spanned by the left-singular vectors of the Jacobian $\nabla_Z g(Z)$ scaled by the corresponding singular vectors. This shows that the diffeomorphic map indeed captures the tangent space of the data manifold to good approximation.

**Adjoint State Method on SE(3):** the repurposing method has the advantage that it is rather general. However, for flow matching and diffusion models, the gradients can be written in terms of a particular adjoint state ODE. For the translational part, this is well known Chen et al. (2018) and given by

$$\frac{d\mathcal{L}(g(Z, z))}{dz} = a_0 \tag{20}$$

where the adjoint state $a_0 \in \mathbb{R}^3$ is obtained by solving the following terminal value problem

$$\frac{da_\tau}{d\tau} = -\sum_{i=1}^{3} a_\tau^i \frac{dv_\theta^i(Z_\tau, z_\tau)}{dz_\tau}\,, \qquad\qquad a_1 = \frac{d\mathcal{L}(R,t)}{dt}\,, \qquad (21)$$

with $\tau \in [0,1]$. In the appendix, we derive a generalization of the adjoint state method for $SO(3)$:

**Theorem 3.** *The Riemannian gradient of the reparameterized loss function*

$$\nabla_Z \mathcal{L}(g(Z,z)) = A_0 \qquad (22)$$

*can be obtained by the following terminal value problem for the Lie-algebra-valued adjoint state $A_\tau = \sum_{i=1}^{3} A_\tau^i T^i \in \mathfrak{so}(3)$:*

$$\frac{dA_\tau}{d\tau} = [V_\theta(Z_\tau, t_\tau), A_\tau] - \sum_{i=1}^{3} A_\tau^i \nabla_{Z_\tau} V_\theta^i(Z_\tau, t_\tau)\,, \qquad A_1 = \nabla_R \mathcal{L}(R,t)\,, \qquad (23)$$

*where $\tau \in [0,1]$ and $[A,B] = AB - BA$ is the commutator of matrices $A$ and $B$.*

*Proof.* See Appendix C.3. $\qquad\qquad\qquad\qquad\qquad\qquad\qquad\qquad\qquad\qquad\qquad\square$

Wang et al. (2024) used the adjoint method on $SO(3)$ to calculate gradients with respect to deterministic optimal control which they used as guidance. As shown in the appendix, this result also holds for any matrix Lie group. A notable advantage of the adjoint state method is that it turns the gradient calculation into a terminal value problem and thus can benefit from modern ODE solvers. However, the adjoint state method has the downside that it assumes that the forward pass has no discretization errors. Depending on the stiffness of the ODE, one may therefore accumulate a systematic error by the discretization effects of both the forward ODE equation 18 and the backward adjoint state ODE equation 23. In practice, we recommend relying on numerical experiments to benchmark the relative performance of the adjoint state method or the repurposed autograd method for the problem at hand. We will provide a modular integration library that implements both methods.

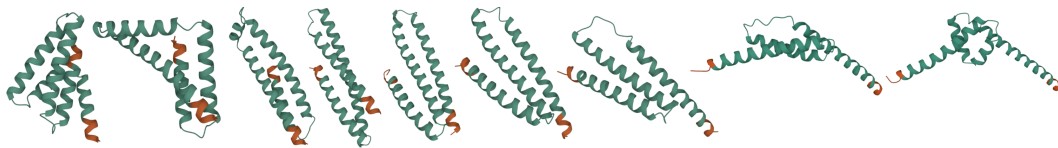

Figure 4: We show various snapshots of the trajectory of diffeomorphic maximization of the distance between the termini of the proteins (marked in red). The optimization traverses a number of plausible conformations.

## 5 EXPERIMENTS

We demonstrate the generality of diffeomorphic optimization in several numerical experiments. We refer to Appendix H for more details.

**$SO(3)$ Manifold:** In Figure 3, we visualize diffeomorphic optimization for a toy dataset on $SO(3)$. Specifically, we create a dataset of S-shape in the angle-axis vectors of the rotations. We then train a flow matching model to sample $SO(3)$ elements on this manifold. The diffeomorphic optimization aims to push the sample as close as possible to the point marked by a star on the S-shaped data manifold. As the figure demonstrates, diffeomorphic optimization leads to a gradient descent trajectory on the data manifold. The green plane is spanned by the left-singular vectors of the Jacobian $\nabla_Z g(Z)$ in the angle-axis representation for a given point $Z \in SO(3)$ scaled by the corresponding singular values which span the tangent space $T_Z SO(3)$ to good approximation. This illustrates that diffeomorphic optimization can harness the differential geometric notions learned by the flow model.

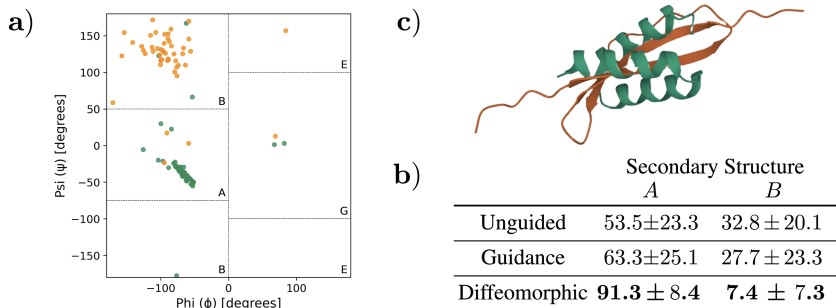

Figure 5: Diffeomorphic optimization is used to change the secondary structure of the protein. For this, we use the ABEGO classification shown on the left. We start from the green protein shown on the top right whose residues are mostly clustered in the A region of the Ramachandran plot which corresponds to $\alpha$-helices. We then optimize the structure such that these residues are pushed to the B region corresponding to $\beta$-sheets. The resulting protein structure is shown in orange in the top right. The final and initial Ramachandran plot are shown on the left. The lower right shows comparison to guidance for 50 samples obtained from the model. Diffeomorphic optimization significantly outperforms guidance despite careful tuning of the baseline hyperparameters described in the appendix.

**Peptide Design:** we compare diffeomorphic optimization to the OC-Flow framework Wang et al. (2024) in which a control parameter $\theta_t$ is added to the flow in the form of $g(z) = z + \int_0^1 v_\theta(x_\tau) + \theta_\tau d\tau$, see Appendix B for more details. We kept the experimental setup identical to Wang et al. (2024) and compare to our proposed diffeomorphic optimization of both translation and rotations of the peptide's backbone. We use the repurposing approach and summarize the results in Table 1. We stress that this improvement can be obtained while being $2\times$ faster in terms of runtime.

Table 1: Evaluation of OC-Flow peptide design.

|  | MadraX ↓ | RMSD ↓ | SSR % ↑ | BSR % ↑ | Stability ↓ | Affinity ↓ | Diversity ↑ |
|---|---|---|---|---|---|---|---|
| Ground-truth | -0.588 | – | – | – | -84.893 | -36.063 | – |
| PepFlow | -0.195 | 1.645 | 0.794 | 0.874 | -45.660 | -26.538 | 0.310 |
| OC-Flow(trans) | -0.229 | 1.774 | 0.797 | 0.876 | -48.380 | -27.328 | 0.323 |
| OC-Flow(rot) | -0.221 | 1.643 | 0.794 | 0.872 | -48.636 | -27.211 | 0.310 |
| OC-Flow(trans+rot) | -0.263 | 2.127 | **0.797** | 0.869 | -48.853 | -27.468 | 0.338 |
| DiffeoOpt | **-0.309** | **1.605** | 0.796 | **0.881** | **-49.417** | **-28.409** | **0.340** |

**Secondary Structure Modification with FrameFlow:** we use FrameFlow Yim et al. (2023a), a state-of-the-art backbone generation flow matching model, to optimize the backbone structure with respect to a given cost function. Figure 4 illustrates an optimization trajectory for which the distance between the protein termini is maximized. This demonstrates that our method can optimize a given protein structure while staying on the data manifold. Similarly, Figure 5 shows that we can also optimize the secondary structure of a protein as specified by the ABEGO classification scheme Wintjens et al. (1996); Kim et al. (2009).
For this, we minimize an energy function that is based on the Ramachandran-PAAPP-combined term in the Rosetta energy Alford et al. (2017); Leaver-Fay et al. (2025) which interpolates energies derived from statistics of backbone dihedral preferences in the PDB on a toroidal grid. The grid values are modified such that the undesired regions are disfavored. We compare to guidance with carefully tuned hyperparameters. Diffeomorphic optimization significantly outperforms the while maintaining designability with results in Appendix H.2.

**Protein-Ligand Docking with DiffDock:** DiffDock is a diffusion model that samples both the center of mass translation and rotation (i.e. an $SE(3)$ element) and the torsion angles of the ligand for a given protein Corso et al. (2023). The model is trained on the PDBBind dataset Wang et al. (2005), which contains experimentally determined protein-ligand docking

structures. We then apply diffeomorphic optimization to maximize the popular VinaSF score of the protein-ligand complex, utilizing its OpenDock implementation Hu et al. (2024).

To enable backpropagation through the solver, we modify the DiffDock generation process to use probability ODE sampling through our custom integration library. We then optimize the score function on pdb test set samples. We compare diffeomorphic optimization to iid sampling followed by selecting the sample with the best docking score using the same computational budget. For this, each gradient descent step is counted as three sampling trajectories, which is quite generous for the baseline. Figure 7 shows the improvement $S_{\text{diffeo}} - S_{\text{iid}}$ obtained by diffeomorphic optimization which significantly outperforms this baseline. Our method can thus efficiently combine physics-based dockingr scores with trained generative models. Further analysis is provided in Appendix H.3.

**Minimization of Rosetta Energy with AlphaFlow:** we consider the Rosetta energy function Alford et al. (2017) which is widely used in the protein community using the tmol pytorch implementation Leaver-Fay et al. (2025) of beta_nov2016_cart. We select the same pdb test set as in the AlphaFlow publication Jing et al. (2024).

This reference proposed several flows that are obtained from protein structure prediction models. We select the ESMFold model for convenience, as it does not require MSA processing. We compare diffeomorphic optimization to the state-of-the-art Rosetta Relax protocol. From our theoretical analysis, we expect that diffeomorphic optimization will lead to better mode mixing. Once an energetically favorable mode is reached, it is cheaper to use standard relaxation to find its minimum. We thus first minimize the Rosetta energy function in the base space of the flow which is then followed by standard Rosetta Relax.

Figure 6 shows that diffeomorphic optimization leads to substantially lower energy values throughout the pdb test set. Furthermore, the energy values of the baseline Rosetta Relax do not improve significantly by running a longer optimization or using more seeds. Thus, the relative advantage is not a function of the higher cost of diffeomorphic optimization. Further analysis is provided in Appendix H.4.

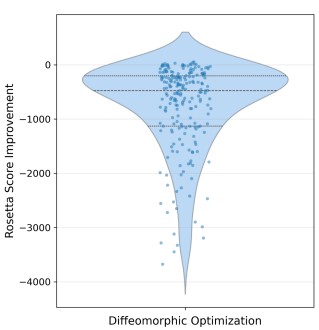

Figure 6: Diffeomorphic optimization applied to AlphaFlow exhibits strong improvement as measured in the Rosetta energy score.

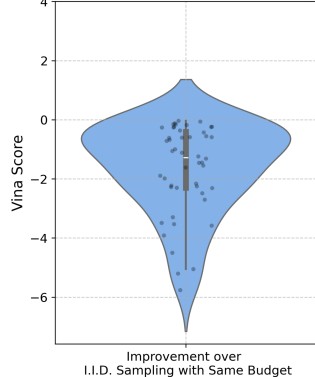

Figure 7: Diffeomorphic optimization leads to better/lower vina score than iid sampling with the same budget.

## 6 CONCLUSION AND LIMITATIONS

We have proposed diffeomorphic optimization which allows us to minimize arbitrary differentiable cost functions on the data manifold. A downside of this approach are the considerable numerical costs due to the backpropagation through the generation. However, sampling costs are of lower concern in protein design where the main bottleneck is experimental wet-lab verification of the designs. It is completely standard in this setting to sample thousands of designs, rank them, and submit only a handful for wetlab experiments. Diffeomorphic optimization provides a more targeted approach to obtain high quality samples.

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

# A    Riemannian Gradient Implementation for $SO(3)$

As derived in Theorem 2, we can efficiently modify existing autograd engines to calculate the $SO(3)$ Riemannian gradient. Specifically, we can wrap the tensor by an operation that acts like the identity in the forward pass and adjusts the gradient to coincide with the Riemannian gradient in the backward pass using Theorem 2. In PyTorch, this can most conveniently be implemented with a backward hook as shown below. We emphasize that this implementation is completely general and ensures that the Riemannian gradient is seamlessly integrated in existing autograd functionality.

```python
import torch
from scipy.spatial.transform import Rotation

def so3tensor(*args, **kwargs):
    x = torch.tensor(*args, **kwargs)

    def so3_backward_hook(grad):
        so3_grad = grad @ x.transpose(-1, -2)
        return so3_grad - so3_grad.transpose(-1, -2)

    if x.requires_grad:
        x.register_hook(so3_backward_hook)
    return x

x = so3tensor(Rotation.random(1).as_matrix(), requires_grad=True)
y = so3tensor(Rotation.random(1).as_matrix())

loss = (x - y).pow(2).sum()
so3_grad = torch.autograd.grad(loss, x)[0]
print(so3_grad)

>>tensor([[[ 0.0000,   0.8490,   0.1187],
           [-0.8490,   0.0000,  -2.2494],
           [-0.1187,   2.2494,   0.0000]]], dtype=torch.float64)
```

# B    Expanded Related Work

**Backpropagation through folding models:** protein hallucination is a version of computational protein design which uses backpropagation through a folding model to its input sequence Anishchenko et al. (2021); Kosugi & Ohue (2022); Goverde et al. (2023); Pacesa et al. (2024). Interestingly, even the most recent work Cho et al. (2025) based on AlphaFold3 avoids explicit backpropagation through its diffusion structure module and instead relies on the pair representation - a limitation that could potentially be overcome by our proposed methods.

**Gradient descent and normalizing flows** has been explored in the explainability literature to generate counterfactual explanations Joshi et al. (2019); Dombrowski et al. (2021; 2023); Dhurandhar et al. (2018) although not for flow matching and diffusion models. Ben-Hamu et al. (2024) proposes to differentiate through flows for controlled generation. Our work builds on this reference by generalizing it to matrix Lie groups, which is of high relevance for proteins. We also provide a detailed theoretical analysis of the method and derive an efficient adjoint state method. Wang et al. (2024) explores related ideas for matrix groups in the framework of optimal control. Specifically, the authors propose to add an additive control to the vector field of the flow. Our approach does not use a control but rather optimizes the initial condition following Ben-Hamu et al. (2024). We discuss the relationship to this reference in more detail in the appendix and compare in detailed numerical experiments to their approach. Liu et al. (2023) similarly relies on control variables but directly applies gradient descent to them. This reference does however not consider matrix groups.

**Adjoint state method on manifolds:** the adjoint state method on manifolds has been discussed in other works - however only in terms of charts Lou et al. (2020); Mathieu & Nickel

(2020) or particular manifolds Rezende & Mohamed (2020); Bacchio et al. (2023); Albergo & Vanden-Eijnden (2022). The adjoint state method derived in our work is applicable to any matrix Lie group and is particularly efficient for the group $SE(3)$ which plays a crucial role the frame-based protein backbone representation.

**Guidance:** a widely used method to bias diffusion and flow-matching models towards certain desiderata is guidance. There exist various flavors of it, such as classifier-based guidance Dhariwal & Nichol (2021), classifier-free guidance Ho & Salimans (2022), and universal guidance Bansal et al. (2023). There has also been substantial recent interest in biasing the generation with additional reward functions Wu et al. (2023); Trippe et al. (2022); Cardoso et al. (2023); Dou & Song (2024); Singhal et al. (2025) often relying on sequential Monte Carlo. However, these methods struggle for guidance potentials that are sensitive to fine-grained details of the final sample, such as force fields or certain biochemical properties, as they tend to rely on few-shot denoising and require carefully hyperparameter finetuing of their starting point and time-dependent weighting factor. We discuss the limitations of guidance in more detail in Appendix F.

### B.1 RELATION TO WANG ET AL. (2005)

A closely related reference is Wang et al. (2005) which outlines a very nice approach to maximize a terminal reward using deterministic optimal control. Specifically, the authors propose to add an additive control $\theta_\tau$ to the vector field of the flow

$$\frac{dx_\tau}{d\tau} = v_\theta(x_\tau, \tau) \qquad \rightarrow \qquad \frac{dx_\tau}{d\tau} = v_\theta(x_\tau, \tau) + \theta_\tau \tag{24}$$

The authors then propose to optimize the control with respect to a (regularized) loss

$$\mathcal{L}(x_1^\theta) + d(x_1^\theta, x_1^0) + \frac{1}{2} \int_0^1 d\tau \, ||\theta_\tau||^2 \tag{25}$$

where $x_1^\theta$ corresponds to the terminal value under the control $\theta_\tau$ and $x_1^0$ is the terminal value for no control. Furthermore, $d(\cdot, \cdot)$ denotes a distance.

A change in initial condition $x_0 \rightarrow x_0 + \delta x_0$ can only be interpreted as a control if the latter is not a function but rather a distribution, i.e. is highly degenerate. Specifically, the control generically needs to be proportional to a Dirac distribution $\delta(\tau)$ as can be seen as follows

$$x_1 = x_0 + \delta x_0 + \int_0^1 v(x_\tau) d\tau = x_0 + \int_0^1 (v_\theta(x_\tau) + \delta(\tau)\delta x_0) \, d\tau \tag{26}$$

and thus implying a distributional control $\theta_\tau = \delta(\tau)\delta x_0$. Note that this distributional control has highly undesirable properties from a numerics standpoint, i.e., it is necessarily divergent for $\tau = 0$ and vanishing for all other values of flow time $\tau$.

## C PROOFS

### C.1 PROOF OF THEOREM 1

**Lemma 1.** *For $g : M \rightarrow N$, it holds that*

$$g(\exp_p(\lambda v)) = \exp_{g(p)}(\lambda \, dg_p(v) + \mathcal{O}(\lambda^2)) \tag{27}$$

*for any $p \in M$, $v \in T_pM$, and small $\lambda \in \mathbb{R}$.*

*Proof.* For proving this statement, it is useful to consider normal coordinates. For $p \in M$ and a neighborhood $U$ of $0 \in T_pM$, these coordinates are given by the chart $\psi : U \subset T_pM \rightarrow M$ with $\psi(v) = \exp_p(v)$. The coordinates $y^\alpha$ of a point $q \in M$ in this chart are obtained by

$$q = \exp_p(y^\alpha e_\alpha) \tag{28}$$

where $e_\alpha$ is an orthonormal basis of $T_p M$ and we have used the Einstein summation convention. This implies that

$$y^\alpha(\exp_q(v)) = v^\alpha \tag{29}$$

Using normal coordinates for both $M$ and $N$, we can expand the function $g$ using Taylor's theorem

$$g(\widehat{\exp_p(\lambda v)}) = g^\alpha(y^\beta) = g^\alpha(\lambda v^\beta) = g^\alpha(0) + \lambda \frac{\partial g^\alpha}{\partial y^\beta} v^\beta + \mathcal{O}(\lambda^2) = \widehat{\exp_{g(p)}}(\lambda \widehat{dg_p}(v) + \mathcal{O}(\lambda^2)) \tag{30}$$

where we have used that the connection $\Gamma$ vanishes in normal coordinates and the hat symbol denotes the coordindate representation. Since the left and the right hand side are written in terms of covariant objects, the result stated in the theorem follows. $\qquad\square$

**Lemma 2.** *Let $g : Z \to X$ be a diffeomorphism with $Z$ being a Riemannian manifold with metric $G$. Then, it holds that*

$$dg_z(grad_z^G \mathcal{L} \circ g) = grad_{g(z)}^{\tilde{G}} \mathcal{L} \tag{31}$$

*where $\tilde{G} = g_* G$ is the pushforward metric of $G$.*

*Proof.* This statement is easily shown in coordinates. Let $z^\mu$ denote coordinates on $Z$ and $x^\alpha = g^\alpha(z)$. Then, the coordinate representation of the pushforward metric $\tilde{G}_{\alpha\beta}$ follows by

$$G_{\mu\nu} dz^\mu dz^\nu = G_{\mu\nu} \frac{\partial z^\mu}{\partial x^\alpha} \frac{\partial z^\nu}{\partial x^\beta} dx^\alpha dx^\beta = \tilde{G}_{\alpha\beta} dx^\alpha dx^\beta \tag{32}$$

In these coordinates, $dg_z(\mathrm{grad}_z^G \mathcal{L} \circ g)$ is given by

$$\frac{\partial x^\alpha}{\partial z^\mu} G^{\mu\nu} \frac{\partial(\mathcal{L} \circ g)}{\partial z^\nu} = \frac{\partial x^\alpha}{\partial z^\mu} G^{\mu\nu} \frac{\partial x^\sigma}{\partial z^\nu} \frac{\partial \mathcal{L}}{\partial x^\sigma} = \tilde{G}^{\alpha\sigma} \frac{\partial \mathcal{L}}{\partial x^\sigma} \tag{33}$$

which corresponds to the coordinate representation of the right hand side. $\qquad\square$

Theorem 1 follows almost immediately using the the two lemmas:

*Proof.* By Lemma 1, it follows that

$$g(\exp_z(-\lambda \,\mathrm{grad}_z^G \mathcal{L} \circ g)) = \exp_{g(z)}(-\lambda \, dg_z(\mathrm{grad}_z^G \mathcal{L}) + \mathcal{O}(\lambda^2)) \,. \tag{34}$$

The right hand side can then be rewritten by Lemma 2 as

$$\exp_{g(z)}(-\lambda \, dg_z(\mathrm{grad}_z^G \mathcal{L}) + \mathcal{O}(\lambda^2)) = \exp_{g(z)}(-\lambda \,\mathrm{grad}_{g(z)}^{\tilde{G}} \mathcal{L} + \mathcal{O}(\lambda^2)) \tag{35}$$

which shows the stated result. $\qquad\square$

## C.2 PROOF OF THEOREM 2

We consider the definition of the Riemannian gradient on the Lie algebra

$$\partial^a f(R) = \frac{d}{d\tau} f(\exp(\tau T^a) R)|_{\tau=0} \tag{36}$$

$$= \frac{df}{dR_{\alpha\beta}} \frac{d(\exp(\tau T^a) R)_{\alpha\beta}}{d\tau}|_{\tau=0} \tag{37}$$

$$= \frac{df}{dR_{\alpha\beta}} (T^a R)_{\alpha\beta} \tag{38}$$

$$= \frac{df}{dR_{\alpha\beta}} T^a_{\alpha\sigma} R_{\sigma\beta} \tag{39}$$

$$= \frac{df}{dR_{\alpha\beta}} T^a_{\alpha\sigma} (R^T)_{\beta\sigma} \qquad\qquad | \quad T^a = -(T^a)^T \tag{40}$$

$$= -\frac{df}{dR_{\alpha\beta}} T^a_{\sigma\alpha} (R^T)_{\beta\sigma} \tag{41}$$

$$= -\text{tr}\left( T^a \frac{df}{dR} R^T \right) \tag{42}$$

$$= -\frac{1}{2}\text{tr}\left( T^a\, 2\frac{df}{dR} R^T \right) \tag{43}$$

$$= \langle T^a, 2\frac{df}{dR} R^T \rangle \tag{44}$$

where we have used the Einstein summation convention and the definition of inner product $\langle A, B \rangle = \frac{1}{2}\text{tr} A^T B$ for Lie algebra elements $A, B \in \mathfrak{so}(3)$. Recall that the Riemannian gradient is given by

$$\partial f \equiv \sum_a T^a \partial_a f(R) \tag{45}$$

Due to the orthonormality of the generators, $\langle T^a, T^b \rangle = \delta^{ab}$ and denoting the antisymmetric part $[M]_A = \frac{1}{2}(M - M^T)$ of a matrix $M$, it thus holds that

$$\partial f = 2\left[ \frac{df}{dR} R^T \right]_A, \tag{46}$$

The antisymmetric component arises in this step because for an arbitrary matrix $M$, we have $\langle T^a, M \rangle = \langle T^a, [M]_A \rangle$.

## C.3 PROOF OF THEOREM 3

The adjoint state is given by

$$A_\tau = \nabla_{Z_\tau} \mathcal{L} \tag{47}$$

for which we want to derive a differential equation for its time evolution.

For this, we define the time evolution operator

$$T_\epsilon : SO(3) \to SO(3), \qquad\qquad Z_\tau \mapsto Z_{\tau+\epsilon} = \exp(\epsilon V_\theta(Z_\tau)) Z_\tau. \tag{48}$$

We use the chain rule equation 91 to derive that

$$A_\tau^a = \nabla_{Z_\tau}^a \mathcal{L} = \sum_b \nabla_{Z_{\tau+\epsilon}}^b \mathcal{L}\, \mathfrak{D}^{ba} T_\epsilon \tag{49}$$

$$= \sum_b A_{\tau+\epsilon}^b\, \mathfrak{D}^{ba} T_\epsilon. \tag{50}$$

The differential equation 90 of the time evolution operator is given by:

$$\mathfrak{D}^{ba} T_\epsilon = \langle T^b, \nabla_{Z_\tau}^a T_\epsilon\, T_\epsilon^\top \rangle \tag{51}$$

$$= \langle T^b, \nabla_{Z_\tau}^a (e^{\epsilon V_\theta(Z_\tau)} Z_\tau)\, Z_\tau^\top e^{-\epsilon V_\theta(Z_\tau)} \rangle, \tag{52}$$

where we have used that $V_\theta \in \mathfrak{so}(3)$ is antisymmetric. We now expand this expression up to quadratic order in the step size $\epsilon$ to obtain

$$\mathfrak{D}^{ba}T_\epsilon = \langle T^b, \nabla^a_{Z_\tau}\left(\left(\mathbb{I} + \epsilon V_\theta(Z_\tau)\right)Z_\tau\right)Z_\tau^\top\left(\mathbb{I} - \epsilon V_\theta(Z_\tau)\right)\rangle + \mathcal{O}(\epsilon^2) \tag{53}$$

$$= \delta^{ba} + \epsilon\langle T^b, [V_\theta, T^a]\rangle + \epsilon\langle T^b, \nabla^a_{Z_\tau}V_\theta(Z_\tau)\rangle + \mathcal{O}(\epsilon^2) \tag{54}$$

We now use the fact that

$$\langle T^b, [V_\theta, T^a]\rangle = -\frac{1}{2}\text{tr}\,T^b\left(V_\theta T^a - T^a V_\theta\right) = -\frac{1}{2}\text{tr}\,T^a\left(T^b V_\theta - V_\theta T^b\right) = \langle T^a, [T^b, V_\theta]\rangle. \tag{55}$$

Expanding $V_\theta = \sum_c V_\theta^c T^c$ and similarly for $[T^b, V_\theta] \in \mathfrak{so}(3)$, we then obtain that the differential is given by

$$\mathfrak{D}^{ba}T_\epsilon = \delta^{ba} - \epsilon[V_\theta, T^b]^a + \epsilon\nabla^a_{Z_\tau}V_\theta(Z_\tau)^b + \mathcal{O}(\epsilon^2) \tag{56}$$

Plugging this result into the chain rule for the adjoint state equation 50, we obtain

$$A_\tau^a = A_{\tau+\epsilon}^a - \epsilon[V_\theta, A_{\tau+\epsilon}]^a + \epsilon\sum_b A_{\tau+\epsilon}^b\,\nabla^a_{Z_\tau}V_\theta(Z_\tau)^b + \mathcal{O}(\epsilon^2)\,. \tag{57}$$

We now rearrange this to isolate the finite time difference on the right-hand-side

$$\tfrac{1}{\epsilon}\left(A_{\tau+\epsilon}^a - A_\tau^a\right) = [V_\theta, A_{\tau+\epsilon}]^a - \sum_b A_{\tau+\epsilon}^b\,\nabla^a_{Z_\tau}V_\theta(Z_\tau)^b + \mathcal{O}(\epsilon)\,. \tag{58}$$

Taking the limit $\epsilon \to 0$ gives

$$\frac{d}{d\tau}A_\tau^a = [V_\theta, A_\tau]^a - \sum_b A_\tau^b\,\nabla^a_{Z_\tau}V_\theta(Z_\tau)^b\,. \tag{59}$$

Multiplying by the generator $T^a$ and summing over the index $a$, we then obtain:

$$\frac{d}{d\tau}A_\tau = [V_\theta, A_\tau] - \sum_b A_\tau^b\,\nabla_{Z_\tau}V_\theta(Z_\tau)^b\,, \tag{60}$$

which is the claimed time-evolution equation of the theorem.

# D   Lightning Review of Differential Geometry and Lie Groups

**Right multiplication:**   Lie groups are smooth manifolds endowed by an additional group multiplication. There is therefore a natural diffeomorphism given by right multiplication

$$R_g : G \to G\,, \qquad\qquad R_g : h \mapsto hg\,, \tag{61}$$

for $h, g \in G$. Let us denote the unit element of the group by $e$. Since right multiplication $R_g$ is a diffeomorphism, its differential

$$(dR_g)_e : T_eG \to T_gG$$

is a linear isomorphism (linear bijective map). We can therefore uniquely identify tangent vectors $v \in T_gG$ and Lie algebra elements $\tilde{v} \in \mathfrak{g} \simeq T_eG$.

**Matrix groups and right multiplication:**   For matrix groups, the differential $(dR_g)_e : T_eG \to T_gG$ takes a particularly simple form

$$(dR_g)_e v = vg \tag{62}$$

where we take the standard matrix product of the Lie algebra element $v$ and the group element $g$ on the right hand side. This statement can be easily checked by noticing that the differential acts on any function $f : G \to \mathbb{R}$ by

$$[(dR_g)_e v]f = \frac{d}{dt}f(\underbrace{\gamma_v(t)g}_{\equiv\tilde{\gamma}(t)})\Big|_{t=0}\,, \tag{63}$$

where $\gamma_v$ denotes the curve associated with $v \in T_eG$, i.e. $\gamma_v(0) = e$ and $\frac{d}{dt}\gamma_v(t)|_{t=0} = v$. As a result, the curve $\tilde{\gamma}$ obeys

$$\tilde{\gamma}(0) = g\,, \qquad\qquad \frac{d}{dt}\tilde{\gamma}(t)|_{t=0} = \frac{d}{dt}\gamma(t)|_{t=0}\,g = vg\,, \tag{64}$$

and is therefore a curve associated with $vg \in T_gG$ as claimed.

**Killing form:**   we first define the adjoint map

$$\text{ad}_u(v) = [u, v] \tag{65}$$

where $u, v \in \mathfrak{g}$. The adjoint map is linear and a Lie algebra homomorphism

$$\text{ad}_{[u,v]} = [\text{ad}_u, \text{ad}_v] . \tag{66}$$

Then the Killing form is given by

$$B(u, v) = \text{Tr}(\text{ad}_u \circ \text{ad}_v) \tag{67}$$

for $u, v \in \mathfrak{g}$ is a bilinear symmetric form that is non-degenerate if the group is semi-simple and compact. We can use the Killing form to define a inner product for $T_e G$ by

$$\langle u, v \rangle_e = -B(u, v) . \tag{68}$$

The negative sign ensures positive definiteness.

**Riemannian metric:**   Often it is more convenient to work with the Lie algebra instead of the various tangent spaces. For example, an inner product $\langle \cdot, \cdot \rangle_e$ on the Lie algebra induces a Riemannian inner product on the group $G$ by

$$\langle v, w \rangle_g \equiv \langle (dR_{g^{-1}})_g v, (dR_{g^{-1}})_g w \rangle_e = \langle vg^{-1}, wg^{-1} \rangle_e , \qquad \text{for } v, w \in T_g G, \tag{69}$$

where we have used that $dR_{g^{-1}} = (dR_g)^{-1}$ to pull tangent vectors back to the Lie algebra. Using $(dR_g)^{-1} = dR_{g^{-1}}$, it immediately follows that the inner product is right-invariant, i.e.,

$$\langle (dR_g)_e \xi, (dR_g)_e \eta \rangle_g = \langle \xi, \eta \rangle_e \tag{70}$$

for Lie algebra elements $\xi, \eta \in T_e G$. For $SO(3)$, we choose antisymmetric generators $T^a$ that are orthonormal with respect to the inner product induced by the Killing form $\langle T^a, T^b \rangle = \frac{1}{2}\text{tr}(T^a)^\top T^b = \delta^{ab}$.

**Riemannian gradient:**   Recall that the Riemannian gradient $\text{grad}_g f \in T_g G$ is the the unique element of the tangent space which obeys

$$\langle \text{grad}_g f, v \rangle_g = df_g\, v \tag{71}$$

for all $v \in T_g G$. Due to the isomorphism between the tangent space $T_g G$ and the Lie algebra $\mathfrak{g} \simeq T_e G$, the Riemannian gradient $\text{grad}_g f \in T_g G$ uniquely corresponds to the Lie algebra element

$$\nabla f \equiv (dR_{g^{-1}})_e\, \text{grad}_g f \in \mathfrak{g} . \tag{72}$$

It is often more convenient to work with this Lie algebra representative of the Riemannian gradient. We will now derive a simple expression for this Lie algebra gradient. By definition of the Riemannian gradient, it holds that

$$\langle \text{grad}_g f, (dR_g)_e v \rangle_g = df_g\left( (dR_g)_e v \right) , \tag{73}$$

for any $v \in \mathfrak{g}$. We can invert equation 72 to obtain

$$\text{grad}_g f = (dR_g)_e \nabla f \tag{74}$$

Using this, we can rewrite equation 73 as follows

$$\langle (dR_g)_e \nabla f, (dR_g)_e v \rangle_g = \langle \nabla f, v \rangle_e , \tag{75}$$

where we have used the right-invariance of the metric equation 70. We thus conclude that

$$\langle \nabla f, v \rangle_e = df_g\left( (dR_g)_e v \right) \tag{76}$$

By definition of the right hand side, it therefore holds for a matrix group that

$$\langle \nabla f, v \rangle_e = df_g(vg) = \frac{d}{dt} f(\exp(tv)g)\Big|_{t=0} . \tag{77}$$

We can expand the Lie algebra representative in terms of generators $\nabla f = \sum_a T^a \nabla^a f$ whose components are given by

$$\nabla^a f = \langle \nabla f, T^a \rangle_e = \frac{d}{dt} f(\exp(tT^a)g)\Big|_{t=0} , \tag{78}$$

where we have used equation 77 and assumed that the generators are chosen to be orthonormal, i.e., $\langle T^a, T^b \rangle_e = \delta^{ab}$. For brevity, we typically refer to the Lie algebra representative of the Riemannian gradient $\nabla f$ as simply the Riemannian gradient in the main part of the paper.

**Exponential map:** For a Lie group, the exponential map takes the form

$$\exp_g(v) = \exp((dR_{g^{-1}})_g v)\, g \tag{79}$$

for $g \in G$ and $v \in T_g G$ where $\exp : \mathfrak{g} \to G$ is the matrix exponential for matrix Lie groups. We note that, by the Lie algebra expression of the Riemannian gradient equation 74, this implies that gradient descent on a Lie group with learning rate $\lambda \in \mathbb{R}$ and loss $\mathcal{L} : G \to \mathbb{R}$ is given by

$$g^{i+1} = \exp(-\lambda \nabla \mathcal{L}) g^i\,. \tag{80}$$

This result is used extensively in the main part of the paper.

**Differential:** Right multiplication $R_g : G \to G$ induces a isomorphism between the tangent spaces $T_g G$ and the Lie algebra $\mathfrak{g}$. Consider the differential $df_g : T_g G \to T_{f(g)} G$ of the map $f : G \to G$. Using the right multiplicative isomorphism equation 62, we can write any tangent space element in terms of a Lie algebra element

$$T_g G \ni \Omega G \leftrightarrow \omega \in \mathfrak{g} \tag{81}$$

Specifically, we can define a Lie algebra representative of the differential

$$\mathfrak{D}f_g \equiv d(R_{f(g)^{-1}} \circ f_g \circ R_g)_e : T_e G \simeq \mathfrak{g} \to T_e G \simeq \mathfrak{g} \tag{82}$$

Using the fact that the right isomorphism amounts to right multiplication equation 62, we can derive a explicit expression for this representative

$$\mathfrak{D}f_g(\omega) = df_g(\omega g) f(g)^\top \tag{83}$$

$$= \frac{d}{dt} f(\exp(t\omega)g)\big|_{t=0} f(g)^\top \tag{84}$$

$$= \frac{df}{dg}(\omega g) f(g)^\top \tag{85}$$

$$= \sum_a \omega_a \frac{df}{dg}(T^a g) f(g)^\top \tag{86}$$

$$= \sum_a \omega_a \frac{d}{dt} f(\exp(t T^a)g)\big|_{t=0} f(g)^\top \tag{87}$$

$$= \sum_a \omega_a \nabla^a f(g)\, f(g)^\top\,. \tag{88}$$

Since $\mathfrak{D}f_g(\omega) \in \mathfrak{g}$, we can expand it as $\mathfrak{D}f_g(\omega) = \sum_b T^b \mathfrak{D}^b f_g(\omega)$. Hence, it holds that

$$\mathfrak{D}^b f_g(\omega) = \sum_a \langle T^b, \nabla^a f(g)\, f(g)^\top \rangle \omega_a \equiv \sum_a \mathfrak{D}f_g^{ba} \omega_a\,, \tag{89}$$

with

$$\mathfrak{D}f_g^{ba} \equiv \langle T^b, \nabla^a f(g)\, f(g)^\top \rangle\,. \tag{90}$$

We can think of $\mathfrak{D}f_g^{ba}$ as being the matrix representation of the differential in the generator basis $T^a$.

**Chain rule:** Let's also consider a function $F : G \to \mathbb{R}$ and its composition $F \circ f : G \to \mathbb{R}$. Then it holds that

$$\nabla^a (F \circ f)_g = \sum_b \nabla^b F_{f(g)}\, \mathfrak{D}f_g^{ba}\,. \tag{91}$$

This follows directly from

$$F \circ f \circ R_g = F \circ R_g \circ R_{g^{-1}} \circ f \circ R_g\,. \tag{92}$$

Taking the differential of this and using the standard chain rule of the differential then gives

$$d(F \circ R_{f(g)} \circ R_{f(g)^{-1}} \circ f \circ R_g)_e = d(F \circ R_{f(g)})_e\, d(R_{f(g)^{-1}} \circ f \circ R_g)_e = \nabla F_{f(g)}\, \mathfrak{D}f_g\,. \tag{93}$$

# E    SO(3) Conventions

Any group element $g \in SO(3)$ can be written as

$$g = \exp(A) \tag{94}$$

with $A \in \mathfrak{so}(3)$ taking value in the Lie algebra

$$\mathfrak{so}(3) = \{A^\mathsf{T} = -A \,|\, A \in \mathbb{R}^{3,3}\}\,. \tag{95}$$

A Lie algebra is, in particular, a vector space. We will choose the basis $T^a$ with $a \in \{1, 2, 3\}$ with

$$T^1 = \begin{bmatrix} 0 & 0 & 0 \\ 0 & 0 & -1 \\ 0 & 1 & 0 \end{bmatrix}, \qquad T^2 = \begin{bmatrix} 0 & 0 & 1 \\ 0 & 0 & 0 \\ -1 & 0 & 0 \end{bmatrix}, \qquad T^3 = \begin{bmatrix} 0 & -1 & 0 \\ 1 & 0 & 0 \\ 0 & 0 & 0 \end{bmatrix}\,. \tag{96}$$

In the context of Lie theory, the basis vectors $T^a$ are also referred to as the generators of the Lie algebra. In particular, any antisymmetric three-by-three matrix can be written as a linear combination of the generators. Furthermore, the generators obey the following commutation relations

$$[T^a, T^b] = \sum_{c=1}^{3} \epsilon^{abc} T^c\,, \tag{97}$$

where $\epsilon^{abc}$ denotes Levi-Civita symbol. It can be checked that

$$\mathrm{Tr}(T^a) = 0\,, \tag{98}$$

$$\mathrm{Tr}(T^a T^b) = -2\delta^{ab}\,, \tag{99}$$

where $\delta^{ab}$ is the Kronecker symbol. Using equation 99, we can equip the Lie algebra with an inner product

$$\langle A, B \rangle = \frac{1}{2} \mathrm{Tr}(A^\mathsf{T} B) = \sum_{a=1}^{3} A^a B^a\,, \tag{100}$$

where we have used that we can express an arbitrary element $A$ of the Lie algebra $\mathfrak{so}(3)$ as $A = \sum_{a=1}^{3} A^a T^a$. Note that our basis $T^a$ is orthonormal with respect to this inner product.

# F    Limitations of Guidance

Guidance is a technique to bias the generation process, i.e., instead of sampling from the model distribution $p$, one aims to sample from

$$p_{\text{tilted}}(x) = p(x) \frac{1}{Z_{\text{guide}}} e^{-\beta E_{\text{guide}}(x)} \tag{101}$$

where $E_{\text{guide}}$ denotes the guidance energy of interest. For score based models, this amounts to a modification of the score

$$\nabla_x \log p_{\text{tilted}}(x) = \nabla_x \log p(x) - \beta \nabla_x E_{\text{guide}}(x)\,. \tag{102}$$

A challenging aspect of guidance is diffusion involves a one-parameter family of density which is typically parameterized in terms of diffusion time $t$ or noise level $\sigma(t)$. For many guidance energies, such as the Rosetta energy function, it is highly non-trivial to calculate the score of the guidance density at time $t$,

$$p_{\text{guide}}(x_t) = \frac{1}{Z_{\text{guide}}} \int p(x_0|x_t) e^{-\beta E_{\text{guide}}(x_0)} dx_0\,, \tag{103}$$

as it involves an untractable marginalization over the unnoisy datasample $x_0$. Here, $p(x_0|x_t)$ denotes the backward kernel, i.e. the density of obtaining the unnoisy data sample $x_0$ when integrating the reverse SDE starting from $x_t$ at diffusion time $t$. The fundamental challenge

of all guidance schemes is to approximate the above marginalization in a suitable manner for the application at hand. This approximation becomes particularly challenging in the limit of $\beta \to \infty$ which corresponds to sampling minimizers of the guidance energy $E_{\text{guide}}$. In this limit, the integrand is heavily dominated by the minimizers (and its neighborhoods for very large but finite $\beta$ as used in numerical experiments). As such, one has to estimate the conditional density $p(x^*|x_t)$ for the minimizers $x^*$ with high precision. For this reason, guidance cannot be expected to work well for minimizing generic guidance energies.

Let us consider the simpler case in which we do not want to minimize the guidance energy, i.e., we do not consider the challenging $\beta \to \infty$ limit. In this case, there are various strategies to approximate the marginalization. None of them are perfect, and each comes with a certain set of tradeoffs. We will only focus on methods which work for a pretrained unconditional model as this is the relevant setting for diffeomorphic optimization. In principle, one could use Monte-Carlo to estimate the guided score

$$\nabla_{x_t} \log p_{\text{guide}}(x_t) \approx \nabla_{x_t} \log \frac{1}{N} \sum_{i=1}^{N} e^{-\beta E_{\text{guide}}(x_0^{(i)}(x_t))} \tag{104}$$

with $x_0^{(i)} \sim p(x_0|x_t)$. For this, we have to repeatedly integrate the reverse SDE starting from $x_t$ and backpropagate through it. There are methods for refined backpropagation through SDEs, such as suitable generalizations of the adjoint state methodKidger et al. (2021); Li et al. (2020) but we are not aware of any work in the protein space applying these techniques. This is possibly due to the considerable technical difficulty, such as the appearance of Stratonovich stochastic integrals, as well as scaling concerns to larger problem sizes. Furthermore, one would need to derive generalizations of these methods for matrix Lie groups.

Universal guidance Bansal et al. (2023) is a technique inspired by classifier guidance Dhariwal & Nichol (2021). In this setting the marginalization is approximated by a zeroth-order saddle point approximation around the expectation value $\hat{x}_0(x_t) = \mathbb{E}[x_0|x_t]$ of the kernel $p(x_0|x_t)$, i.e.,

$$p_{\text{guide}}(x_t) \approx \frac{1}{Z_{\text{guide}}} \int \delta(\hat{x}_0(x_t) - x_0) \, e^{-\beta E_{\text{guide}}(x_0)} dx_0 = \frac{1}{Z_{\text{guide}}} e^{-\beta E_{\text{guide}}(\hat{x}_0(x_t))}, \tag{105}$$

which can be a good approximation at low noise levels (small diffusion time) but tends to be very poor at high noise. To alleviate this problem, guidance is often combined with a heuristic schedule which switches the guidance score on when the diffusion time is lower than some threshold (often in an adiabatic manner). However, this results in a fundamental tension: we want the guidance signal to kick in as early as possible in the generation process to meaningfully change the selected mode of the sample. On the other hand, we do not want to start the guidance too early as the zeroth-order saddle point approximation would be very poor. Whether such a delicate balance can be struck often heavily depends on the problem setting and requires careful (and costly) hyperparameter tuning. As a general rule of thumb, energy functions that depend only on the coarse grain details of the sample $x_0$ seem to be easier to deal with. Examples include the classification of image types or the sentiment of a sentence. However, energy functions in physics (which are highly relevant in the protein space) are typically not of this type. For example, the Rosetta energy is extremely sensitive to the repulsive component of the atomic interactions resulting in numerically destabilizing energy values even if the global fold and secondary structure of the predicted unnoisy protein is roughly correct.

Recently, there has been considerable interest in combining diffusion models with Sequential Monte Carlo (SMC) to facilitate sampling from the tilted distribution $p_{\text{tilted}}$. We base our discussion on the very recent Singhal et al. (2025) which introduces Feynman-Kac steering unifying a number of previous approaches in a common theoretical framework. The basic idea is to modify the transition kernel $p(x_{t-1}|x_t)$ of the reverse diffusion

$$p(x_t|x_{t+1}) \to p(x_t|x_{t+1})G(x_1, \ldots, x_t) \tag{106}$$

by a guidance potential $G$ which fullfills the consistency condition

$$\prod_{t=1}^{0} G(x_1, \ldots, x_t) = \frac{1}{Z_{\text{guide}}} e^{-\beta E_{\text{guide}}(x_0)} . \tag{107}$$

This ensures that the joint distribution of the sampling trajectory $(x_1, \ldots, x_t, \ldots x_0)$ is given by

$$p_{\text{guide}}(x_1, \ldots, x_0) = p(x_1, \ldots, x_0) \frac{1}{Z_{\text{guide}}} e^{-\beta E_{\text{guide}}(x_0)} \tag{108}$$

and thus has the desired marginal $p_{\text{guide}}(x_0)$. In principle, any guidance potential satisfying the consistency condition will lead to asymptotic gurantees but in practice a careful choice of potential is vital to avoid prohibitive variance. This is because, at each step of the reverse diffusion one uses importance sampling to reweight the sample $x_t$ according to the tilted transition kernel equation 106. In practice, one often has to choose guidance potentials $G$ that involve challenging marginalization. For example, a widely used guidance potential is the expected guidance energy $E_{guide}$. Thus this approach faces analogous challenges to universal guidance, particularly in the $\beta \to \infty$ limit.

Another important point of differentiation between guidance and diffeomorphic optimization is that the latter is particularly suited in applications in which we want to improve a particular sample $x_0$ with respect to some cost function. For guidance, we would need to incorporate a constraint in the guidance potential that ensures similarity to $x_0$. For diffeomorphic optimization, we perform gradient descent updates that iteratively refine the sample.

# G    Complexity and Efficiency of Gradient Estimations through Flows

A forward pass evaluates the network layer by layer, applying linear transformations and nonlinearities to propagate an input through the model. Its cost is determined by the size and number of layers, with matrix multiplications dominating the total FLOPs. A general backward pass is more expensive because it must compute gradients with respect to all parameters. This involves propagating gradients backward through each layer and forming gradient tensors for every weight matrix in each layer and the layers input for further backward propagation. These extra operations roughly double the compute relative to the forward pass, and they also require storing all intermediate activations, which increases memory consumption and can become a limiting factor on modern hardware.

A backward pass that computes gradients only with respect to the input is cheaper. It is comparable to a single backward pass as the parameter gradients do not have to be computed. The computational effort becomes close to that of the forward pass, with only a small overhead for the backward sweep.

Gradient checkpointing reduces memory consumption during backpropagation by storing only a subset of intermediate activations from the forward pass and recomputing the missing ones when needed. In standard backpropagation, all activations must be kept in memory because each layer's gradient depends on its forward output; this creates a memory footprint that grows linearly with depth and often becomes the bottleneck in training large models. Gradient checkpointing breaks the computation graph into segments, saves only the boundary activations, and discards the rest. During the backward pass, the discarded activations are recomputed on the fly by running partial forward passes within each segment. This trades additional compute for a substantial reduction in memory usage. The memory cost can drop from linear to roughly the square root of the number of layers in optimal schemes, enabling deeper networks or larger batch sizes on the same hardware. The extra compute cost is typically a factor of $1.5$–$2\times$, but the trade-off is advantageous when memory is the limiting resource. Modern implementations generalize this idea through customizable checkpoint policies, selective recomputation, and integration with automatic differentiation frameworks, making gradient checkpointing a standard technique in large-scale training, especially for transformer architectures and diffusion models where activation tensors dominate memory load.

ODE solvers compute trajectories by applying numerical integration steps, with cost driven by repeated evaluations of the vector field. Higher-order or adaptive methods improve accuracy but increase per-step compute, so total complexity depends on both the solver order and the number of required steps. In neural ODEs, most of the expense comes from evaluating the neural network that defines the dynamics. Using the autograd engine to compute gradients through a ODE solution keeps the entire computational graph in memory, resulting in $\mathcal{O}(NL)$ scaling with a second pass over the graph in reverse direction required for the backward pass with a total cost of $\mathcal{O}(NL)$.

The adjoint method recomputes the state trajectoy and backpropagates the gradients by integrating a separate reverse-time ODE, avoiding storage of intermediate states and keeping memory use low. Compared to gradient checkpointing it does not require storing any values besides the vectorfield as it recomputes both the trajectory and the gradients during the backward pass. This leads to a memory complexity of $\mathcal{O}(L)$ as only the network parameters and their activations need to kept in memory at any time. This comes at the price of additional solver calls and potential numerical instability or even divergence from the trajetory of the forward pass, since the backward integration can amplify errors and often requires tighter tolerances. After $\mathcal{O}(NL)$ steps of compute in the forward pass, the adjoint method recomputes the forward pass to reconstruct the trajectory as well as one backward pass to compute the gradient, resulting in an additional $\mathcal{O}(2NL)$ compute complexity during the backward pass. As a result, adjoint-based gradients are memory-efficient $\mathcal{O}(L)$ but computationally more expensive with $\mathcal{O}(3NL)$. Solving a reverse time ODE comes with potential numerical issues as the recomputed trajectories diverge and subsequently lead to diverging gradients.

Checkpointing provides a middle ground by storing selected solver states and recomputing segments during backpropagation. This reduces memory usage to $S = N/K$ at any given time where $S$ is the segment length. Storing $K$ checkpoints in the form of a state of the ODE is negligible as it amounts to $K$ samples. For $K$ checkpoints, we only have to solve $S$ steps at any one time, while the recomputation between the $K$ checkpoints amounts to going forward and backward on the segments compute complexity of $\mathcal{O}(2KSL) = \mathcal{O}(2NL)$. It balances compute and memory more predictably, making it a practical choice for many neural ODE applications.

To quantitatively measure the performance of both the gradient checkpointing and adjoint method, we evaluated each gradient against the ground truth gradient obtained with numerical differentiation. Figure 10 shows the gradient error over increasing modelling dimensions as a funtion of the step size and the checkpointing interval. For the Euclidean part, we integrated the function $dx_t = \cos(t/2\pi) \cdot x_t$ and for the $SO(3)$ part, we integrated $dR_t = \cos(t/2\pi) \sum_a^3 v_a T^a \theta(v)$ with $v = [1.0, -0.2, 0.1]^T$ and $\theta(v) = \|v\|$. The initial condition was sampled from a Gaussian distribution in Euclidean space and the respective axis-angle space for the $SO(3)$ elements. The dimensionality was determined by the number of $SE(3)$ ODE's integrated in parallel for $t \in [0, 1]$. We used this toy ODE's to establish the numerical behavior of the autograd checkpointing and the adjoint methods for backpropagating gradients through ODE solvers.

The analysis suggests that the adjoint method with a checkpointing at every step yields gradients very close to its autograd gradient. Decreasing the checkpointing frequency requires recomputing longer parts of the trajectory which accumulates errors. Overall a trend emerges that decreasing the step size is beneficial for both adjoint and autograd gradients, and particularly for very small stepsizes, the difference reduces significantely.

We also extended this analysis to the FrameFlow model Yim et al. (2023a) model but due to the computational cost of numerical differentiation for the ground truth gradient refrained from evaluating multiple checkpointing frequencies. For this reason we kept the checkpointing frequency fixed at every five steps. One can observe that the adjoint gradients backpropagated through the $SE(3)$ ODE generally more imprecise than their autograd counterparts. For commonly used stepsizes of $1/25$ to $1/100$, the total deviation can be quite significant, but correcting for the per dimension gradient deviation, the gradient error is small.

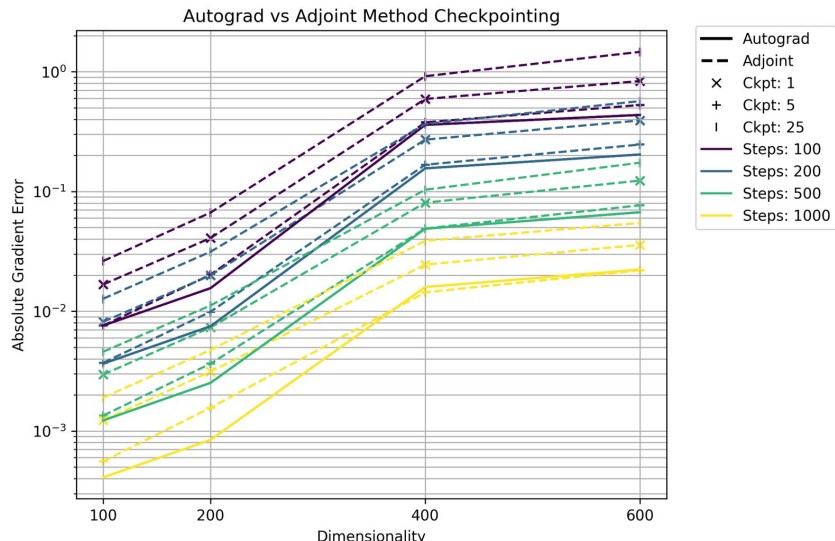

Figure 8: Gradient Error as a function of dimensionality and the number of steps

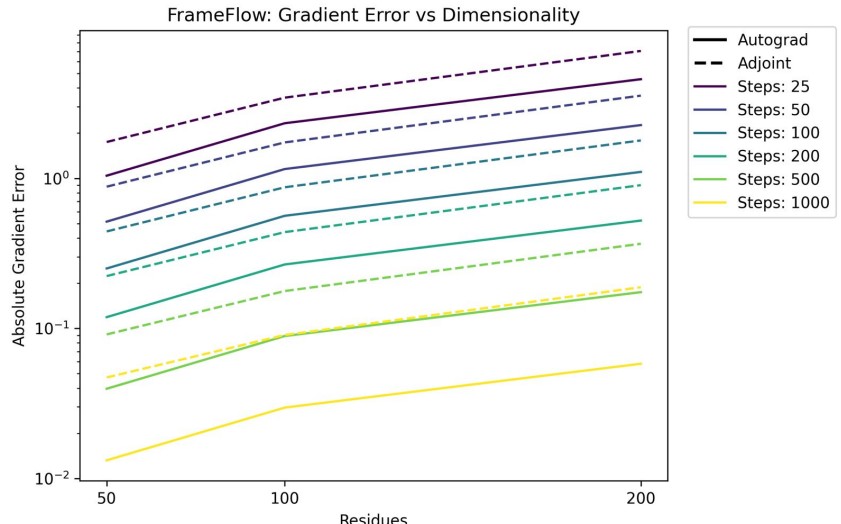

Figure 9: Gradient Error as a function of dimensionality and the number of steps

Finally, we compare the gradient error of the AlphaFlow architecture Jing et al. (2024). Again, adjoint gradient estimators perform worse than their autograd counterparts. It should be noted that AlphaFlow is a flow defined on the $C_\beta$ carbon atoms and does not model a rotation group, compared to the previous two models. We want to note that the evaluation of these gradients were on the order of hours, particularly for the highest step sizes.

## H  EXPERIMENTS

### H.1  SO(3) MANIFOLDS

We generate a manifold in the space of $SO(3)$ matrices. To plot the rotation matrices easily, we parameterize the SO(3) elements in their axis-angle vectors. The manifold in the axis-angle representation was generated with the S-curve function of sklearn, `sklearn.datasets.make_s_curve(n_samples=250000, noise=0.1)`.

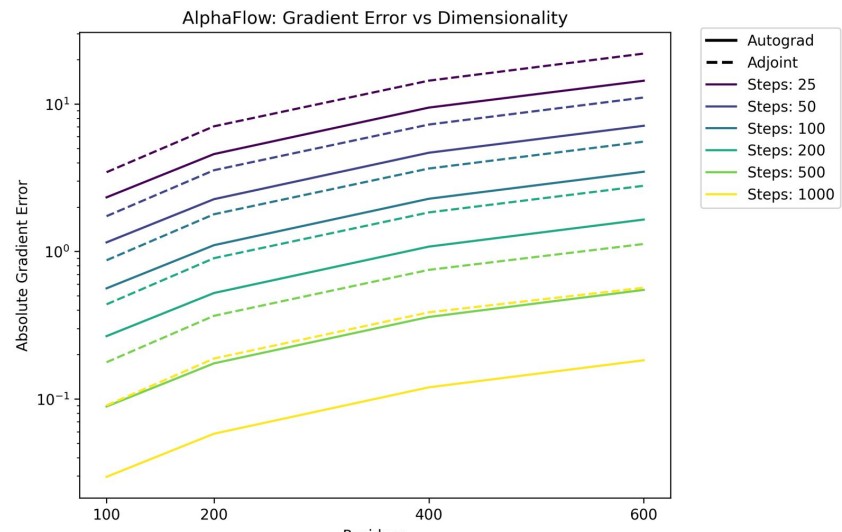

Figure 10: Gradient Error as a function of dimensionality and the number of steps

The flow matching model was parameterized with a four layer MLP with SELU activation functions and 64 hidden units in each of the hidden layers. For training, the model was optimized with Adam using a learning rate of $1e^{-3}$ over 20 epochs on the training data with conditional flow matching to enforce that the predicted flow on the manifold of SO(3) matrices closely matches the target logarithmic map on the manifold. During sampling, the vector field was integrated with the Crouch-Groussmann integrator with a step size of $dt = 1/200$ from 0 to 1.

For the sample optimization, a target sample $z^*$ was experimentally determined and a second random sample was drawn as the initial value $x^{(0)} = g(z^{(0)})$ of the optimization problem. The mean squared error (MSE) between the target sample $x^*$ and the sample $x^{(0)}$ being optimized served as the loss function $\mathcal{L}$. The sample $z^{(i)}$ was then optimized by calculating the gradient $d\mathcal{L}(g(z^{(i)}))/dz^{(i)}$ and performing 20 steps of gradient descent in the $SO(3)$ base space with a base space learning rate of 0.1.

The tangent space was calculated by evaluating the Jacobian matrix $J_g$ with `torch.nn.functional.jacobian` at point $z$ and its corresponding target space point $x = g(z)$. Subsequently, the two largest eigenvalues with the corresponding left columns were computed with a SVD and plotted with a scaling of 1.5 the point $x$.

## H.2  Secondary Structure Optimization

The secondary structure experiments were conducted on top of the FrameFlow codebase Yim et al. (2023a). For training and architectural details, we refer to the manuscript.

We used the model parameters inn the provided model checkpoints by the original authors. The samples $x$ with the time index $t \in [0, 1]$ consist of backbone elements of each residue centered on the Ca carbon atom. Each of the $L$ backbone elements is parameterized by its translation and rotation and are elements of the SE(3) group. The flow generates an element $(t, R)$ of the $SE(3)$ group for each residue. Upon completion of the sampling process, these transformations are applied to the idealized backbone coordinates.

For the experiment described in the main part of the paper, the loss function $\mathcal{L}$ was calculated as a function of the distance between the atom positions in the first and the last residue. The distance loss was implemented by taking the ReLU function over the difference between the desirable target distance and the actual distance, i.e. $\mathcal{L} = \left( \max(0, d^* - d(x_0^{(i)}, x_L^{(i)})) \right)^2$ where the target distance $d^*$ between the atom positions in the two termini $x_0^{(i)}$ and $x_L^{(i)}$ in the

backbone structure of length $L$ at optimization step $(i)$ of the structure was chosen as $d^* = L/2$. The distance metric was chosen as the mean squared error $d(x, y) = \sqrt{\sum_j (x_j - y_j)^2}$ over the atoms $j$ in two residues $x$ and $y$. The order of subtraction $\max(0, d^* - d)$ ensures a loss that becomes zero if $d \geq d^*$ and that the two termini are repelled from each other. We also experimented with an attraction loss in the sense of $\max(0, d - d^*)$ which worked equaly well. In fact, we were able to dynamically switch losses during optimization and the termini were pulled apart and together smoothly as determined by the loss with all intermediate optimization states $x^{(i)}$ staying on the data manifold. When switching between attracting and repelling loss function it was beneficial to reset the momenta of the adaptive optimizers like Adam to accelerate the optimization. For sampling we integrated with the Euler integration method with a step size of $dt = 1/25$. The learning rate for the optimization was chosen at a constant 0.5 for 300 optimization steps.

For the optimization of the dihedral angles according to the ABEGO scheme, we utilized a differentiable implementation of Rosetta energy function in PyTorch, i.e. tmol Leaver-Fay et al. (2025). We use a specific component of the full energy function that takes the dihedral angles of neighboring backbone elements as an input. We modified this component such that it increases the contribution for dihedral angles corresponding to $\beta$-sheets.

In more detail, the combined Ramachandran/p_aa_pp energy term in tmol interpolates the energies (the negative-log probabilities) stored in a toroidal-grid lookup-table over phi and psi dihedrals. To push the generated samples away from the alpha-helical region of conformation space, we created our own custom lookup table where we dramatically increased the energies for an elliptical region of the Ramachandran map that covers the alpha-helical region. The energies at the edge of this ellipse are not uniform, so our first crude assignment of energies inside this ellipse that sloped away from a central peak to a particular value at its edge frustrated the minimizer, trapping conformations in the weird minima at the edge of the ellipse. In our second attempt, we instead labeled each grid cell inside the ellipse by the closest cell outside of the ellipse and then interpolated the energies along the distance from the center of the ellipse (to which we assigned an energy of $+20$) to the ellipse's edge. This led to the desired behaviour and we therefore chose this approach. The resulting grid is shown in Figure 11.

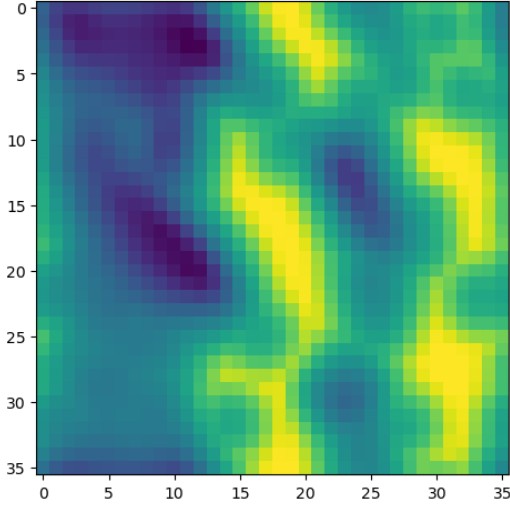

Figure 11: Grid used for Ramachandran guidance.

Three important metrics in machine learning aided protein design are *designability*, *diversity* and *novelty*. We consider designability the most pertinent to our task as diversity and novelty are of lesser interest when trying to converge on a single optimal structure. The optimization towards a single structure makes diversity collapse as expected. Similarly, since our aim is not to generate diverse samples but rather optimize towards a singular structure, novelty is a

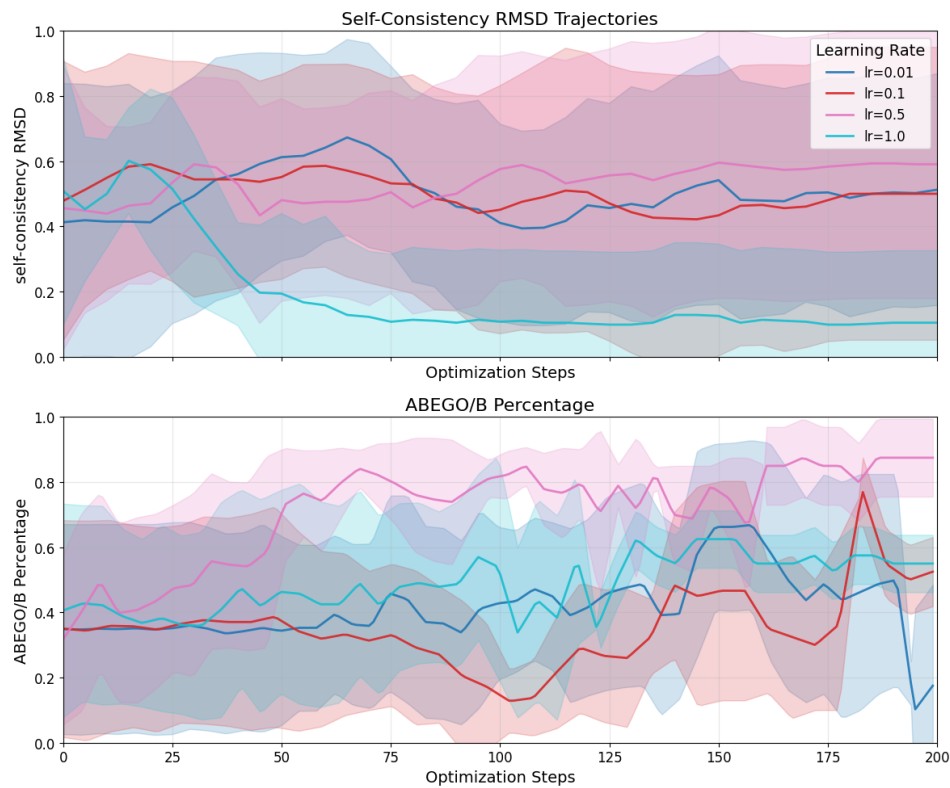

Figure 12: Self-consistent RMSD and the optimized ABEGO percentage over the course of diffeomorphic optimization.

misplaced metric in this case. Empirically, the designability of the optimization trajectories agree well with the designability of the initial values drawn from the generative model. Only for larger learning rates of 1.0 does the designability experience a significant drop while at the same time only marginally improving the percentage of dihedral angles classified as $B$ in the ABEGO scheme.

Since the FrameFlow model does not predict side chains, we interpreted all residues as alanine. We applied the sampled translations and rotations to the idealized backbone atom14 positions to obtain the sampled backbone geometry. The score function was minimized with gradient descent in the base space with a learning rate of 0.1 for 500 steps.

We implemented several versions of guidance to obtain maximally competitive baseline results. Specifically, we implemented both a single step and and a multi step denoiser combined with the loss function described above. The guidance was started at $t = 0.1$. We selected the starting point of the guidance signal by hyperparameter line search over $t = 0.8, 0.1, 0.25, 0.5, 0.75$. We also found that it was benefical to stop sampling early at $t = 0.98$ to prevent distortive numerical pathologies that arose due to the divisor of the optimal transport transport path close to $t = 1$. The guidance vector field was obtained by evaluating the loss on the atom14 positions of each backbone residue and computing the gradient with respect to the corresponding element in the base space. For weighting the guidance term, we tested the theoretically motivated weighting function $b_t$ from Lipman et al. (2024) but obtained better results with a constant, time-independent weight of $w = 25$. This value was selected using a line-search hyperparameter sweep for $w = 0.1, 1, 10, 25, 50, 100, 1000$. For evaluating the guidance term on a sample $x_t$, we tested both taking the direct gradient of the loss function, $\partial L(x)|_{x=g(x_t)}/\partial x$, and backpropagating back through the denoiser $\partial L(g(x_t))/\partial x_t$. We chose the latter as it yielded better results. Experimentally, we also found that only using the translation term (and not the rotational part) as guidance improved the performance. This may be due to the fact that on $SO(3)$, there is no equivalence between score and denoising as in Euclidean space. We furthermore carefully ablated the number of denoiser steps used in multistep denoising. This is because fair comparison to diffeomorphic optimization should allow for a generous budget of denoising steps as the method is numerically costly. As shown in Table 2, even significant increase in the denoising steps does not lead to a pronounced gain in performance.

To demonstrate that the numerical cost of diffeomorphic gradient calculations can be reduced by coarse-grained backpropagation through the solver, we performed the following ablation study: we measured the performance of diffeomorphic optimization as a function of the number of steps used for sampling and calculating gradients during optimization. To this end, we ran diffeomorphic optimization with $10, 25, 50, 75$ and $100$ integration steps during optimization, and evaluated the optimized sample after optimization with $1000$ integration steps. The results are shown in Figure 13. Compared to guidance, even a very small number of steps used for sampling during optimization yields good performance. For this experiment, we measured the performance over 10 seeds as compared to 50 seeds in the main part.

### H.3 Docking Optimization

For docking optimization we used the DiffDock code base. This diffusion model samples from the manifold of permissible rotational degrees of freedom of a small molecule on $SO(2)$ and its global translation and rotation on $SE(3)$. In order to make the sampling process differentiable, we implemented the probability flow formulation of the diffusion models which provided a deterministic map between base space and target space. For a differentiable loss, we considered two different metrics: the confidence head of the diffusion model and the OpenDock scoring function Hu et al. (2024).

Opendock is a physics-inspired scoring function and implements the Vina Score in a differentiable manner in PyTorch. One of the technical challenges of incorporating OpenDock as a suitable loss function were posed by its reliance on PDBQT file formats. In addition to the positional information stored in the PDB file format, the PDBQT file format additionally stores the partial charges, torsional bonds, and autodock atom types. To work around this constraint, we first saved the ligand conformations generated by DiffDock to disk in

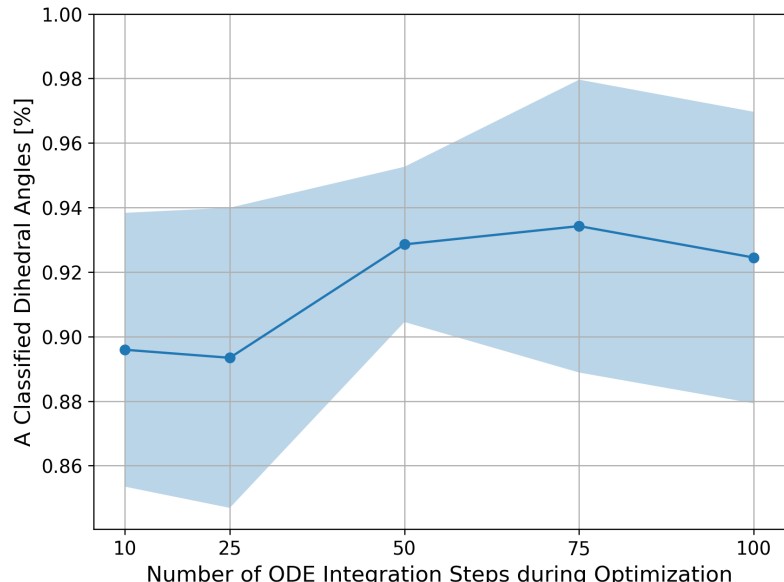

Figure 13: The performance of diffeomorphic optimization (higher is better) as a function of the number of ODE integration steps taken for each evaluation during optimization. After optimization, we evaluated all optimized samples with a 1000 steps. Diffeomorphic optimization is robust even when using just 10 integration steps during optimization.

| | | | Secondary Structure | |
| | | | A | B |
|---|---|---|---|---|
| Unguided | | | $53.5 \pm 23.3$ | $32.8 \pm 20.1$ |
| | ODE Steps | Denoiser Steps | | |
| Guidance | 200 | 1 | $63.3 \pm 25.1$ | $27.7 \pm 23.3$ |
| | | 50 | $63.5 \pm 24.8$ | $27.4 \pm 23.1$ |
| | | 100 | $63.8 \pm 24.5$ | $27.1 \pm 22.9$ |
| | | 200 | $64.1 \pm 24.1$ | $26.8 \pm 22.6$ |
| | 500 | 1 | $64.2 \pm 24.4$ | $26.6 \pm 22.4$ |
| | | 50 | $64.4 \pm 24.0$ | $26.3 \pm 22.1$ |
| | | 100 | $64.7 \pm 23.7$ | $26.0 \pm 21.9$ |
| | | 200 | $65.0 \pm 23.4$ | $25.8 \pm 21.6$ |
| | 1000 | 1 | $65.0 \pm 23.6$ | $25.5 \pm 21.5$ |
| | | 50 | $65.1 \pm 23.4$ | $25.3 \pm 21.3$ |
| | | 100 | $65.2 \pm 23.2$ | $25.1 \pm 21.0$ |
| | | 200 | $65.3 \pm 23.1$ | $24.9 \pm 20.8$ |
| | ODE Steps | Optim Steps | | |
| Diffeomorphic | 25 | 500 | $\mathbf{91.3 \pm 8.4}$ | $\mathbf{7.4 \pm 7.3}$ |

Table 2: The addition of more compute for guidance leads to diminishing returns. Increasing the the number of integration steps both in the flow sampling as well as in the denoiser does not automatically lead to a higher target metric as measured with the percentage of $A$ classified dihedral angles. All guidance terms were weighted with a constant $w = 25$. Diffeomorphic optimization samples with 25 integration steps and applies 500 gradient descent steps in the base space. Diffeomorphic optimization yields higher $A$ classification scores with a substantially lower standard deviation in performance.

the SDF file format, converted it to the PDBQT and reloaded it with the additionally generated partial charges created during the conversion. At this point the computational graph connecting the base space variables to inputs of the differentiable loss scoring function was suspended due to the reading and writing to disk. To reconnect the computational graph with the input of the Opendock scoring function, we substituted the atom positions in the loaded PDBQT data structure with the atom positions originating from the DiffDock model.

The confidence head of the diffusion model is a neural network and was trained to predict whether the generated conformation is within 2 Angstroms of the ground truth. We additionally include on the scoring of OpenDock into our experimental setup which is a differentiable implementation of Vina. Following the original authors recommendation on the confidence prediction, we screened for the entries in the publicly available PDBBind test data set that were assigned a confidence logit of larger than 0 by the DiffDock confidence head. This resulted on a subset of roughly 78 PDBBind ligand-protein pairs on which the results were reported. We perform 10 optimization steps with an Adam-style optimizer with a learning rate of 0.1. Experimentally, we observe that the gradients of the rotation and torsional degrees of freedom play a dominant role in the norm of the total gradient. To counteract this, we introduced separate scalings for the learning rates of the rotation and torsions and multiply the learning rates for these two sets of degrees of freedom by an additional 0.1.

To ensure fair comparison, we draw more samples for the sampling baseline in order to match the computational budget used by diffeomorphic optimization. Since both checkpointing and the adjoint method recompute the activations during the backward pass through the ODE solver, we used a budget corrector of $3\times$ a single sampling step. This budget corrector number could be reduced more with coarser checkpointing and fewer steps, but we observed favorable experimental results even for this setting and thus did not explore this direction further.

## H.4 TERTIARY STRUCTURE OPTIMIZATION

We built on the alphaflow codebaseJing et al. (2024) and chose the ESMFold-based architecture as it does not require costly MSA. We ensured that diffeomorphic optimization works both for distilled and non-distilled case but large-scale testset-wide experiments were conducted with the latter as it leads to significant speed-up. The model ingests the pairwise distances between random atom positions to predict the backbone and the side chain atom positions conditioned on the sequence information of the protein. As a differentiable loss function, we used the tmol implementation of of beta_nov2016_cart Rosetta score function which support pytorch autograd to compute gradients. As the architecture contains non-differentiable distograms, we implemented a differentiable straight-through version thereof harnessing the sigmoid function $\sigma$,

$$d_{soft} = \sigma(\beta \cdot (d - lower)) \cdot (1 - \sigma(\beta \cdot (d - upper)))$$
$$d_{gram} = \text{detach\_grad}(d) + (d_{soft} - \text{detach\_grad}(d_{soft}))$$

where *lower* and *upper* denote the lower and upper bound of the discrete bin in which the distance $d$ should have been placed. While the forward pass is unaffected by the soft distogram implementation, the gradients are affected by the smooth approxiation. Higher values of the $\beta$ temperature result in a steeper sigmoid function at the cost of a stiffer gradient surface.

As a baseline we used the state-of-the-art Rosetta Relax protocol Alford et al. (2017) without any diffeomorphic optimization. For random sidechain repacking, Rosetta Relax uses $n$ seeds which are commonly referred to as structures in the Rosetta community. This is followed by gradient descent in which the repulsive terms in the energy function are adiabatically increased with a linear schedule. Packing and gradient-based minimization is then repeated $k$ times.

In Figure 15 we evaluated the Rosetta Relax baseline with an increasing amount of compute measured as the product of the number of seeds $1, 3, 5$ with the number relaxation cycles $3, 5$. We were not able to discern an improvement compared to diffeomorphic optimization in conjunction with Rosetta Relax.

For diffeomorphic optimization, we used Adam with a learning rate of 0.1 and otherwise standard pytorch hyperparameters. This was then followed by standard Rosetta relax to reduce numerical cost. The performance of this protocol as a function of diffeomorphic optimization steps is shown in Figure 16.

In addition to the experiments reported in the main part of the paper, we also compared the performance of our diffeomorphic Rosetta Relax to a sampling-based baseline. For this baseline, we simply sample from AlphaFlow with the equivalent computational budget used by diffeomorphic optimization. We select the sample with the lowest energy which is then minimized by the same standard Rosetta relax as used in the last step of the diffeomorphic protocol. As shown in Figure **??**, diffeomorphic optimization again outperforms this baseline.

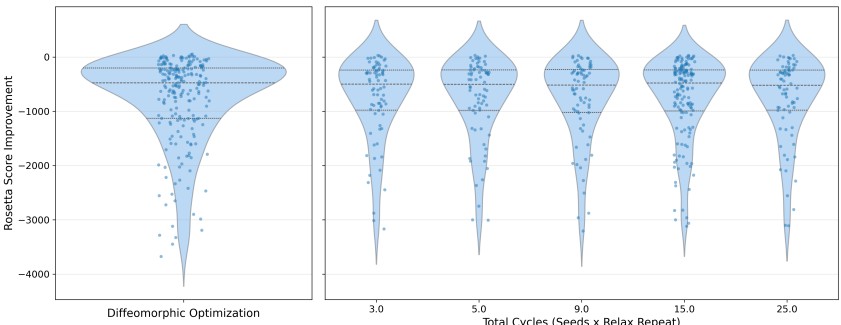

Figure 14: Diffeomorphic optimization of the Rosetta energy function: each point denotes a prediction of ESMFlow for an element of the pdb test set of Alphaflow for which we measure the improvement $E_{\text{diffeo}} - E_{\text{baseline}}$ of diffeomorphic Rosetta Relax over standard Rosetta relax. **Left:** Diffeomorphic Rosetta Relax outperforms Rosetta Relax and improves the Rosetta score with the default Rosetta Relax values of three structures and three repeated relaxations. **Right:** This improvement holds across an increasing computational budget as measured in terms of the total number of relaxations.

## I  ETHICS STATEMENT

This work introduces a general optimization framework for generative models and demonstrates its application to protein design. We emphasize that our experiments are limited to computational settings and publicly available protein data. No human or animal subjects are involved, and no wet-lab synthesis or testing is performed.

The methodology has potential implications for drug discovery and bioengineering. While these applications can provide substantial benefits, such as accelerating the design of therapeutic proteins, they also raise concerns about dual use in harmful contexts. We mitigate these risks by focusing exclusively on methodological contributions, using standard benchmark datasets, and refraining from any direct applications to pathogenic proteins or toxins.

Finally, we adhere to principles of reproducibility and transparency. All experiments are conducted with publicly available datasets, and we plan to release code to ensure accessibility and verifiability of our results.

## J  REPRODUCIBILITY STATEMENT

We have made efforts to ensure the reproducibility of our results. A detailed description of the mathematical foundation for diffeomorphic optimization algorithm is provided in Section 3, with proofs and theoretical derivations included in Appendix C. The experimental protocols, including optimization settings and evaluation metrics, are described in SectionH.

All datasets used in this work are publicly available. We describe the data sources and preprocessing steps in Appendix H. For protein experiments, we use standard benchmark

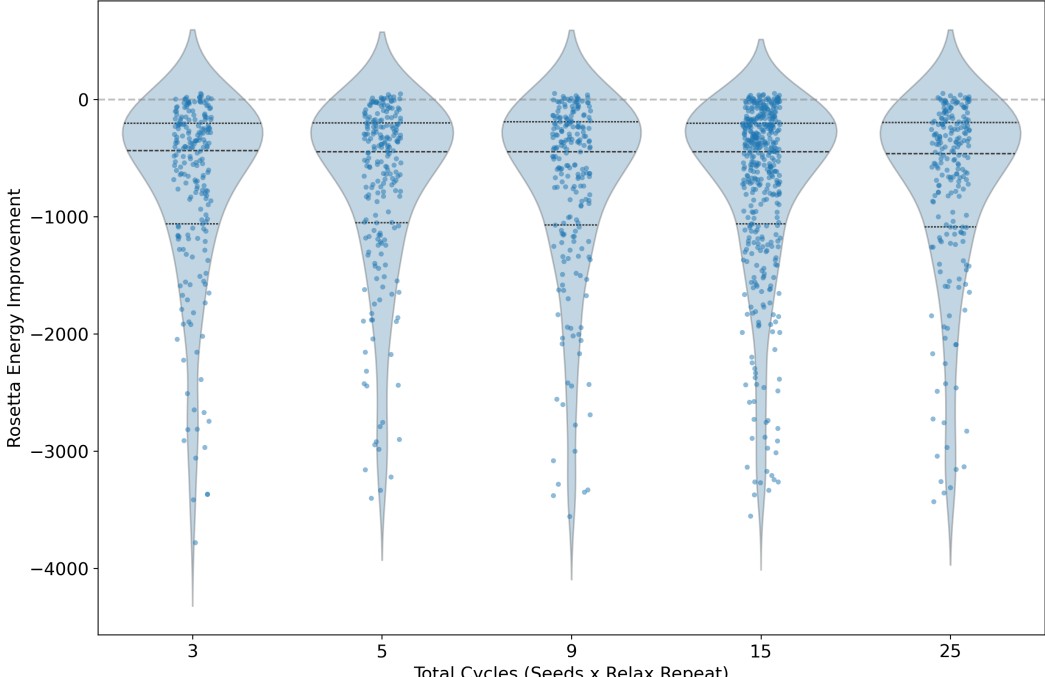

Figure 15: Diffeomorphic optimization in combination with Rosetta Relax yields improved (lower) Rosetta energies when directly compared to Rosetta Relax. This statement holds even with increasing computational budget for the baseline. In this experiment, we varied the number of Rosetta Relax random seeds (referred to as structures in the Rosetta community) and the number of relaxations for each random seed. The product of both Rosetta Relax random seeds and relaxations is plotted on the horizontal axis. We can observe no substantial benefits of providing Rosetta Relax with additional compute budget compared to diffeomorphic optimization which consistently improves the final Rosetta energy.

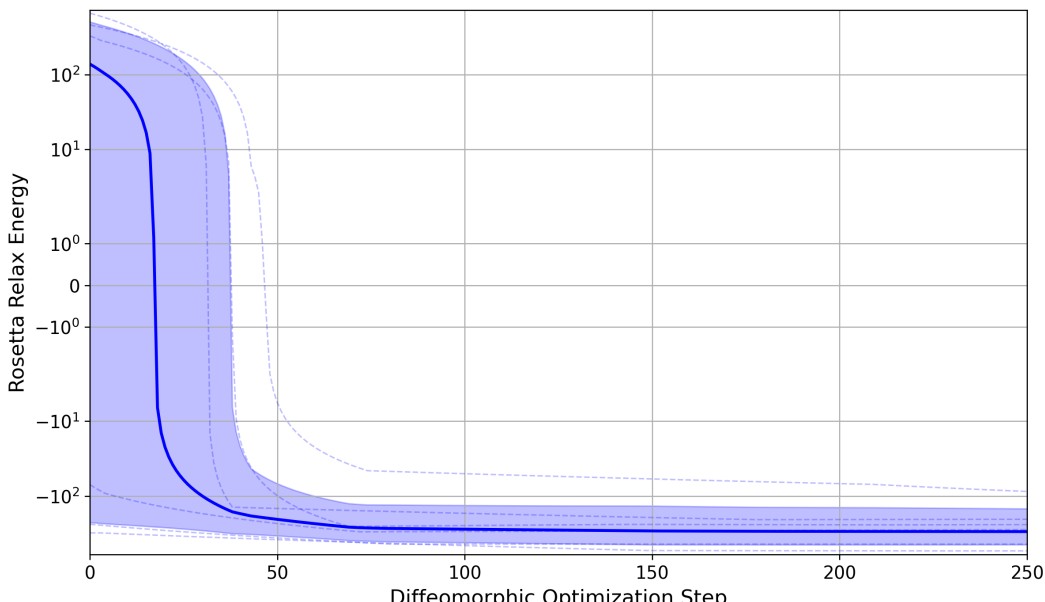

Figure 16: Diffeomorphic Optimization in combination with Rosetta Relax can achieve substantial energy reductions in a small number of steps. This is exemplified by the high energies at the beginning of the optimization curves which corresponds to running Rosetta Relax without a diffeomorphic optimization step which are subsequently is minimized by several orders of magnitudes while remaining stable.

datasets from the Protein Data Bank, and we clearly document filtering criteria and feature representations.

The contributions in this work are of theoretical nature. We will release our custom written integration package for Euclidean and matrix Lie groups upon publication, in the hope that it will spur more research.

## K  Large Language Models

All conceptualization, methodology, code development, experimental analysis, and the creation of figures and results were carried out entirely by the authors. Large language models were not involved in the design of the research, execution of experiments, or interpretation of findings. We used large language models (LLMs) as an aid in polishing the exposition of the manuscript, for example by rephrasing sentences or helping to overcome writer's block. During coding they used useful for refactoring unwieldy research code.

