# OpenReview forum: "Diffeomorphic Optimization"
_ICLR.cc/2026/Conference — Submitted to ICLR 2026_

### Official Review · Reviewer_AExf · 2025-10-26

**Soundness:** 3
**Presentation:** 3
**Contribution:** 3
**Rating:** 8
**Confidence:** 4

**Summary:**

This paper proposes Diffeomorphic Optimization, a novel framework that performs optimization constrained to the data manifold learned by a generative model. The core idea is to reparameterize the optimization problem in the latent base space of a diffeomorphic generative model (e.g., a flow or diffusion model). By leveraging differential geometry, the authors prove that gradient descent in the latent space is first-order equivalent to Riemannian gradient descent on the data manifold. The method is further extended to matrix Lie groups, such as SO(3) and SE(3), enabling applications to 3D geometry problems like protein structure refinement and protein–ligand docking. Empirical results demonstrate faster convergence, smoother optimization trajectories, and improved geometric consistency compared to Euclidean and guidance-based methods.

**Strengths:**

This work makes a significant contribution at the intersection of geometric machine learning and generative model optimization.
- Introduces a new and conceptually elegant framework linking differential geometry and generative model optimization.
The diffeomorphic parameterization idea is both theoretically deep and practically impactful.
- The proofs (Section 3) are mathematically solid, with correct use of Riemannian and Lie group formalism.
- Writing is precise and pedagogical; the paper explains nontrivial geometry intuitively without oversimplifying.
- Provides a general foundation for manifold-aware optimization, relevant to numerous scientific and AI domains.
-  Demonstrates real impact on SE(3) applications (e.g., protein docking) where geometric consistency is crucial.

**Weaknesses:**

- While results on protein-related tasks are convincing, evaluations on non-biological manifolds (e.g., moulecules, crystal materials) would help establish broader generality.
- The theoretical guarantees rely on the diffeomorphism being smooth and bijective. In practice, diffusion models may only approximate this property. A discussion or metric quantifying deviation from diffeomorphism would strengthen the work.

**Questions:**

- How sensitive is the optimization quality to the accuracy of the learned diffeomorphism?
Does the framework degrade gracefully if the generative mapping g slightly violates invertibility or smoothness?
- Have the authors explored second-order extensions (e.g., Riemannian Newton or natural gradient methods) within the same diffeomorphic framework?
- Is the adjoint-state method in Section 4 computationally scalable for large molecule systems (many SE(3) transformations)?
- For reproducibility, will the authors release the code and pretrained diffeomorphic models used in the experiments?

---

> ### Author Response · Authors · 2025-11-24
>
> We would like to thank the reviewer for taking the time to read our manuscript and provide feedback on it. We are more than happy to try to clarify questions and address points raised.
>
> >Weaknesses
> >
> >*While results on protein-related tasks are convincing, evaluations on non-biological manifolds (e.g., moulecules, crystal materials) would help establish broader generality.*
> >
>
> We agree that more experiments are always helpful in determining advantages and breaking points. We value the reviewers proposed modalities as promising future work. Protein design is however a highly active and emergent area of machine learning research. ML-guided design of antibodies and enzymes has the potential to transform important areas such as drug discovery. Last year’s nobel prize was award for protein design. Current approaches often sample tens of thousands of designs and then filter them to select only a handful, see the recent boltzgen paper for an example. Our approach proposes a more targeted approach to obtain high quality samples form a flow or diffusion models.
>
> >*The theoretical guarantees rely on the diffeomorphism being smooth and bijective. In practice, diffusion models may only approximate this property. A discussion or metric quantifying deviation from diffeomorphism would strengthen the work.*>
>
> We fully agree with this statement. We tried to derive some bounds involving the KL divergence between the flow and the true data distribution. However, the bounds are necessarily worst-case and therefore quite loose and we did therefore not include them in the manuscript. Finding a metric that quantifies the deviation in practice would be highly desirable but is unfortunately a very challenging task and an active area of research, see for example “On the Invertibility of Invertible Neural Networks” by Behrmann et al.
>
> >Questions
> >
>
> >*How sensitive is the optimization quality to the accuracy of the learned diffeomorphism? Does the framework degrade gracefully if the generative mapping g slightly violates invertibility or smoothness?*>
>
> Empirically, we have observed graceful behavior. The framework is dependent on the underlying learned diffeomorphism and the used cost function. The Diffdock experiments initially used the confidence head as a loss function, which caused unstable training behavior as the confidence changed with the change in the base space variable. For that reason, we switched to a physical loss function with opendock which resulted in more stable optimization that is also of higher practical relevance.
>
> >*Have the authors explored second-order extensions (e.g., Riemannian Newton or natural gradient methods) within the same diffeomorphic framework?*
> >
> We have not considered Riemannian Newton or natural gradients for this work but these are interesting ideas and we thank the reviewer for pointing them out. Our software package, which we will publish upon acceptance, does implement the Crouch-Grossman solver which is a 3rd order Runge-Kutta solver. The CG solver has the advantage of being applicable to both Euclidean as well as Lie groups and can thus be used for both cases.
>
> >*Is the adjoint-state method in Section 4 computationally scalable for large molecule systems (many SE(3) transformations)?*
> >
> For the Frameflow experiment, we’ve successfully optimized structures up to 200 residues large (200 SE(3) transformations). For the AlphaFlow examples, we’ve optimized structures up to 800 residues large. This is at least one order of magnitude larger than previously proposed flow optimization methods on QM9 (the 9 refers to maximally 9 heavy atoms as the data dimensionality) or peptides. Proteins of this size are of high practical importance, for example in the form of antibodies or enzymes.
>
> >*For reproducibility, will the authors release the code and pretrained diffeomorphic models used in the experiments?*
> >
>
> We will release our code upon acceptance. We will also open source our custom quadrature package which implements ODE integrators for both Euclidean and SO(3) groups. It implements all combinations of [Euclidean, SO(3), SE(3)] x [SE(3)-adjoint, autograd] x [checkpointing] together into a compact code package.
>
> As our method is principally a inference-time method, we rely on the public model checkpoints provided by the authors of the original FrameFlow, DiffDock, AlphaFlow and PepFlow papers. As such, we such the model parameters are not updated within our frameowrk. We minimally modified the sampling processes to be differentiable and subsequently wrapped the sampling process in an relatively standard optimization loop.
>
> We invite the reviewer to read our updated manuscript. As is customary, we've highlighted updated parts in blue.

---

> > ### Comment · Reviewer_AExf · 2025-11-28
> >
> > Thank you for your response. I will keep my original score.

---

> > > ### Author Response · Authors · 2025-11-28
> > >
> > > Thank you for your swift reply and for your valuable feedback.

---

### Official Review · Reviewer_39ZQ · 2025-10-31

**Soundness:** 2
**Presentation:** 2
**Contribution:** 1
**Rating:** 0
**Confidence:** 4

**Summary:**

This paper is based on D-flow and extends gradient-based guidance in SO(3) space to guide protein backbone generation. The demonstrated applications are interesting. However, the contributions are limited in methodology, experiments, and theory (see weakness).

**Strengths:**

- Clear presentation: The paper provides illustrative toy data experiments, algorithm pseudocode, and detailed implementations of geometric operators. The writing is clear and free of obvious errors.
- Interesting domain application: protein backbone secondary structure modification, pocket-ligand docking, and energy optimization are important applications in the field of protein design.

**Weaknesses:**

- From a method perspective, this method extends D-flow to SO(3) space, which has already been explored in previous literature, e.g., [1].
- From an application perspective, the paper lacks quantitative evaluation against existing guided generation methods.
- From a theoretical perspective, the main results (Theorems 1 and 2) are restatements of well-established results in Riemannian optimization and matrix Lie groups (e.g., [2,3]), with limited novel theoretical contribution.

[1] Wang, Luran, et al. "Training free guided flow matching with optimal control." arXiv preprint arXiv:2410.18070 (2024).

[2] Absil, P-A., Robert Mahony, and Rodolphe Sepulchre. Optimization algorithms on matrix manifolds. Princeton University Press, 2008.

[3] Do Carmo, Manfredo Perdigao, and J. Flaherty Francis. Riemannian geometry. Vol. 2. Boston: Birkhäuser, 1992.

**Questions:**

- Complexity and efficiency are not clearly discussed.
- It is unclear how much the proposed method differs from D-flow in practice. If we ignore the rotations (i.e., frame orientations) and only consider Cα atoms in protein backbone generation, which is in SE(3), the method appears to reduce to a standard D-flow formulation. The paper does not explicitly clarify whether any additional benefits arise beyond applying D-flow to Euclidean coordinates embedded in SE(3).

---

> ### Author Response · Authors · 2025-11-24
>
> We thank the reviewer for taking the time to read our manuscript and their valuable feedback.
>
> >*From a method perspective, this method extends D-flow to SO(3) space, which has already been explored in previous literature, e.g., [1].*
> >
> We thank the reviewer for pointing out the interesting and very relevant reference [1]. We have added a detailed discussion of similarities and differences to our approach to the revised manuscript. Specifically, [1] operates in the framework of deterministic optimal control. For this, [1] adds to the vector field of the flow $f(x_t,t)$ an additive term, called control $\theta_t$, i.e.
>  $ \frac{d}{dt} x_t = f(x_t, t) \quad \rightarrow \quad \tfrac{d}{dt} x_t = f(x_t, t) + \theta_t$
>
> This control $\theta_t$ is then optimized to maximize a terminal reward, see their nice Fig 1 for an illustration. Our approach does not rely on guidance in the form of control but rather optimizes the initial condition of the ODE, which results in a numerically similar (in terms of gradient computation) yet conceptually different framework. As we discuss in detail in Appendix B.1 of the revised manuscript, changing the initial condition can only be interpeted an additive control if $\theta_t$ is distributional, i.e. divergent at $t=0 $ but vanishing for all other flow times which is obviously ill suited for numerical optimization.
>
> The authors of [1] derive a backward ODE to calculate gradients with respect to the control $\theta_t$. In contrast, our approach requires the gradient wrt the initial condition. To this end, we derive i) the adjoint state method suitable generalized for SO(3) and ii) a repurposing method of standard autograd. Note that backward equation (34) of [1] involves the Hamiltonian evaluated at the optimal control $H^*$. Our adjoint state backward equation does not involve a control.
>
> We have replicated the highest dimensional experiment (peptide design) in [1] and our method not only significantly outperforms [1], see table below, but also scales to significantly larger system sizes. For example, we optimize the full Rosetta energy function on large proteins of up to ~800 residues - at least one order of magnitude larger than the peptides considered in [1]. Proteins of this size, such as antibodies or enzymes, are of great commercial and scientific interest. To the best of our knowledge, this manuscript is the first to achieve this.
>
> >*From an application perspective, the paper lacks quantitative evaluation against existing guided generation methods.*
> >
>
> We compare in detail to guidance, carefully tuning the baseline by extensive hyperparameter search, see Table 2 in Appendix H.2 (previously Appendix G.2) in the revised manuscript. We have also added a comparison to [1] on their most challenging task, i.e. optimization of a peptide with respect to the MadraX energy. The results are summarized in the following table clearly demonstrating the power of our method. We were able to improve markedly over the proposed method of [1]. We note that we are able to achieve these results 2x speed advantage compared to [1].
>
> | | MadraX ↓ | RMSD ↓ | SSR % ↑ | BSR % ↑ | Stability ↓ | Affinity ↓ | Diversity ↑ |
> |---|---|---|---|---|---|---|---|
> | Ground-truth | -0.588 | – | – | – | -84.893 | -36.063 | – |
> | PepFlow | -0.195 | 1.645 | 0.794 | 0.874 | -45.660 | -26.538 | 0.310 |
> | OC-Flow(trans) | -0.229 | 1.774 | 0.797 | 0.876 | -48.380 | -27.328 | 0.323 |
> | OC-Flow(rot) | -0.221 | 1.643 | 0.794 | 0.872 | -48.636 | -27.211 | 0.310 |
> | OC-Flow(trans+rot) | -0.263 | 2.127 | **0.797** | 0.869 | -48.853 | -27.468 | 0.338 |
> | **DiffeoOpt** | **-0.309** | **1.605** | 0.796 | **0.881** | **-49.417** | **-28.409** | **0.340** |

---

> > ### Author Response · Authors · 2025-11-24
> >
> > > *From a theoretical perspective, the main results (Theorems 1 and 2) are restatements of well-established results in Riemannian optimization and matrix Lie groups (e.g., [2,3]), with limited novel theoretical contribution.*
> > >
> > Theorem 1 shows that diffeomorphic optimization is equivalent to gradient descent on the data manifold up to second order corrections in the learning rate - an evidently valuable result for the machine learning community as it clarifies the relation between the two approaches in a rigorous manner. We could not find this result in the references [2,3]. We would be thankful if the reviewer could point us to them.
> >
> > The value of theorem 2 lies in the fact that it allows to easily repurpose the autograd engine of existing deep neural network libraries with a few lines of code, see Listing 1 in the revised manuscript. Current implementations of the Riemannian gradient rely on bespoke grad implementations which do not interact well with other important components of deep learning libraries, such as optimizers. See, for example, FrameDiff’s Listing 2 (https://arxiv.org/pdf/2302.02277 , page 25) which is widely used throughout the protein generative model literature. In the light of the reviewer’s comments, we will move the code example of Appendix A to the main part directly after the theorem in order to illustrate this point.
> >
> > > Questions:
> > >
> > >*Complexity and efficiency are not clearly discussed.*
> > >
> > We have added a discussion of the complexity of our method to the appendix of the paper. We describe computational cost and efficiency gains possible in ODE solvers and backpropagation through them. Additionally, we ran extensive numerical experiments quantifying the gradient error on toy models, FrameFlow and AlphaFlow comparing the ground truth gradient obtained via numerical differentiation to the gradients obtained through the repurposed autograd and the adjoint method. We invite the reviewer to give us feedback on appendix G.
> >
> > >*It is unclear how much the proposed method differs from D-flow in practice. If we ignore the rotations (i.e., frame orientations) and only consider Cα atoms in protein backbone generation, which is in SE(3), the method appears to reduce to a standard D-flow formulation. The paper does not explicitly clarify whether any additional benefits arise beyond applying D-flow to Euclidean coordinates embedded in SE(3).*
> > >
> > Embedding rotational matrices in euclidian space is well-known to be problematic for optimization. As a concrete example, consider the angle-axis representation. This representation becomes highly degenerate for angle-axis vectors of small norm. Furthermore, composition of rotation matrix can not be achieved by adding the angle-axis vectors. Rather, one has to map them to rotation matrices, then perform the addition, and then take the logarithmic map to infer the resulting angle-axis vector. The logarithmic map is well-known to be prone to numerical instability. We are happy to add an experiment illustrating this if the reviewer thinks that’s useful.
> >
> > We invite the reviewer to our updated manuscript.

---

> > > ### Comment · Reviewer_39ZQ · 2025-11-27
> > >
> > > Your new experiments addressed my major concern about quantitative evaluation.
> > >
> > > I appreciate your extensive additional experiments and clarifications. I will carefully review your updates and will reconsider my score accordingly.
> > >
> > > Thanks

---

> > > > ### Author Response · Authors · 2025-11-28
> > > >
> > > > Thank you for your swift reply. We are looking forward to your feedback. Please let us know if we can further clarify or improve the manuscript.

---

### Official Review · Reviewer_aRBD · 2025-11-01

**Soundness:** 3
**Presentation:** 2
**Contribution:** 3
**Rating:** 6
**Confidence:** 3

**Summary:**

This paper introduces Diffeomorphic Optimization, a novel method for optimizing arbitrary differentiable cost functions on data manifolds by leveraging flow-based and diffusion generative models. The key insight is that these models learn diffeomorphic (smooth and invertible) maps from simple base distributions to complex data distributions, allowing optimization to be performed in the simpler base space rather than directly on the data manifold.  A significant technical contribution is extending this framework to matrix Lie groups (SO(3) and SE(3)), which are crucial for protein structure representation. The authors develop two methods for backpropagation through ODE solvers on these groups: (1) repurposing existing autograd engines to compute Riemannian gradients, and (2) deriving an adjoint state method for matrix Lie groups. The method is demonstrated on protein design tasks using state-of-the-art generative models (FrameFlow, DiffDock, AlphaFlow), showing successful optimization of secondary structure, protein-ligand docking scores, and Rosetta energy functions while maintaining physically plausible structures throughout the optimization trajectory.

**Strengths:**

The paper provides a mathematically rigorous foundation connecting differential geometry, generative models, and optimization. Theorem 1 establishes that gradient descent in base space is equivalent to gradient descent on the data manifold up to quadratic corrections, giving the core theoretical motivation. The two proposed methods for handling backpropagation (autograd repurposing and adjoint state method) are both theoretically sound and practically implementable.

The method demonstrates clear improvements across diverse protein tasks.

**Weaknesses:**

The method requires backpropagation over the entire ODE integration trajectory, which is computationally expensive. While the authors argue this is acceptable in protein design due to wet-lab bottlenecks, it limits broader applicability.

While the authors' optimization scheme allows us to enforce manifold constraints, it is not clear whether it biases the optimization towards certain minima. As the optimization objective is highly non-convex, it is essential to explore and obtain a reasonable estimate of the global minima rather than focusing on gradient descent to find a local minima. Sampling allows us to see a broader range of solutions rather than finding a single optimal solution. Under this light, it is unclear if the proposed methodology provides a practical advantage over guided sampling methods.

The theory and experiments do not provide any details on the accumulation of estimation errors during backpropagation over the entire ODE trajectory. Some analysis of the estimation error would be highly relevant for practical adoption.

**Questions:**

Please refer to the Weaknesses section.

---

> ### Author Response · Authors · 2025-11-24
>
> First of all, we would like to thank the reviewer for reading and understanding our manuscript. We would in the following like to address the points raised.
>
> > *The method requires backpropagation over the entire ODE integration trajectory, which is computationally expensive. While the authors argue this is acceptable in protein design due to wet-lab bottlenecks, it limits broader applicability.*
> >
> We certainly agree that our method is expensive but numerical costs can be alleviated by using a lower number of backward steps. This is a particularly viable strategy during the initial phase of optimization for which a coarser gradient signal is often sufficient. This was recently successfully used in the BioEmu paper (https://www.biorxiv.org/content/10.1101/2024.12.05.626885v1.full.pdf , S.3.6, bottom page 28). We did not explore this direction in depth in our manuscript but it certainly represents a very promising direction of future research. There has also been considerable work on sampling with a low number of steps from flow and diffusion models, see, for example, https://arxiv.org/abs/2303.01469 and https://arxiv.org/abs/2505.13447, which would also reduce the cost of the gradient.
>
> Inference-time compute is increasingly adopted throughout the machine learning literature and known to be quite costly. For flow-matching and diffusion models, diffeomorphic optimization can provide a more targeted approach than simple sampling-based schemes.
>
> > *While the authors' optimization scheme allows us to enforce manifold constraints, it is not clear whether it biases the optimization towards certain minima. As the optimization objective is highly non-convex, it is essential to explore and obtain a reasonable estimate of the global minima rather than focusing on gradient descent to find a local minima. Sampling allows us to see a broader range of solutions rather than finding a single optimal solution. Under this light, it is unclear if the proposed methodology provides a practical advantage over guided sampling methods.*
> >
> The beauty of diffeomorphic optimization lies in the fact that the data distribution becomes unimodal in the base space of the flow. By pulling the gradient back onto the base space, diffeomorphic optimization performs gradient in a space for which the data distribution is unimodal - a very favorable property for performing gradient descent.
>
> We however fully acknowledge that this relies on a well-trained flow. For example, optimization using flows that drop modes of the data distribution will inevitably not be able to probe this part of the optimization space. Similarly, imperfectly trained flows will not be able to fully transform the data distribution into a unimodal one. This represents an inherent limitation of our method.
>
> We compare in detail to guidance, carefully tuning the baseline by extensive hyperparameter search, see Table 2 in Appendix H.2. This analysis clearly demonstrates the benefits of our method. In the revised manuscript, we have also added a comparison to the current state-of-the-art flow optimization method [1] on their most challenging task, i.e. optimization of a peptide with respect to the MadraX energy. Our method not only outperforms this baseline but also scales to proteins which are at least an order of magnitude larger (from 50 residues to ~800 residues) while doing so 2x faster than [1], see Section 5 in the revised manuscript.
>
> Finally, diffeomorphic optimization and sampling can be combined in a straight forward manner. In our experiments on DiffDock, we first generated promising samples via sampling, and subsequently improved the most promising samples with diffeomorphic optimization. Even after adjusting the computational budget to take into account the computational overhead of diffeomorphic optimization, our method still yields decisive improvements on the metrics.
> [1] https://arxiv.org/html/2410.18070v1
>
> > *The theory and experiments do not provide any details on the accumulation of estimation errors during backpropagation over the entire ODE trajectory. Some analysis of the estimation error would be highly relevant for practical adoption.*
> >
> We have added a qualitative discussion and a quantitative analysis of this to Appendix G. There we analyze the gradient estimation error as a functions of the step sizes as well as the checkpointing frequency employed utilized both in gradient checkpointing for autograd gradients and the trajectory recomputation for the adjoint method. We evaluate the gradient error for a toy model, FrameFlow and AlphaFlow. Please let us know if you have any further feedback on how to improve this.

---

> > ### Author Response · Authors · 2025-11-24
> >
> > We have updated the manuscript and highlighted updated parts in blue. We invite the reviewer to read the the updated manuscript.

---

### Meta-Review · Area_Chair_jDmV · 2026-01-07

**Summary:**

This paper aims at optimizing a function defined on data manifold which is implicit. The main idea is to use flow-based and diffusion model to create a diffeomorphism between the data manifold and another manifold embedded in an ambient Euclidean space, and the authors made some very interesting observation that optimization in the coordinates of the latter is empirically easier. The first half of the paper was written in generality, but the second half specializes in only SO(3) and SE(3), which is less implicit. While some reviewer appreciates the mathematical rigor, some other reviewers and I found the mathematical part rather standard. In fact, some reviewer had significant concerns about novelty and technical contributions, and while questions about experiments were potentially resolved during the rebuttal process, those concerns remained. Recognizing the potential of the idea, I encourage the authors to take the discussions into consideration and re-submit a revised version.

**Reviewer Concerns:**

Reviewers may not be fully convinced in my opinion.

**Reviewer Scores:**

Just a guess: they might not increase by too much.

---

### Decision · Program_Chairs · 2026-01-26

Reject